

# Gauging the Kitaev chain

## Umberto Borla[1,2⋆†], Ruben Verresen[3†], Jeet Shah[4] and Sergej Moroz[1,2]

**1** Physik-Department, Technische Universität München, 85748 Garching, Germany
**2** Munich Center for Quantum Science and Technology (MCQST), 80799 München, Germany
**3** Department of Physics, Harvard University, Cambridge, Massachusetts 02138, USA
**4** Indian Institute of Science, Bangalore, 560012, India

⋆ umberto.borla@tum.de
† Both authors contributed equally to this work.

## Abstract

We gauge the fermion parity symmetry of the Kitaev chain. While the bulk of the model becomes an Ising chain of gauge-invariant spins in a tilted field, near the boundaries the global fermion parity symmetry survives gauging, leading to local gauge-invariant Majorana operators. In the absence of vortices, the Higgs phase exhibits fermionic symmetry-protected topological (SPT) order distinct from the Kitaev chain. Moreover, the deconfined phase can be stable even in the presence of vortices. We also undertake a comprehensive study of a gently gauged model which interpolates between the ordinary and gauged Kitaev chains. This showcases rich quantum criticality and illuminates the topological nature of the Higgs phase. Even in the absence of superconducting terms, gauging leads to an SPT phase which is intrinsically gapless due to an emergent anomaly.



# 1  Introduction

Gauging a global symmetry is one of the most fruitful concepts in modern physics. Underpinning general relativity [1] and the Standard Model of particle physics [2], gauge theories also ubiquitously emerge in many-body quantum systems. They are indispensable for understanding fractionalized quasiparticles, deconfined phases, topological order and exotic quantum critical points [3–5].

Decades of research on gauge theories have unveiled a rich phenomenology. Nevertheless, certain aspects are still fertile ground for exploration. For instance, gauge theories are not fully understood in the presence of boundaries [6–12]—being intimately related to the equally subtle issue of defining entanglement entropy [13–18]—and in this work we will gain some new insights into this matter. Secondly, while it is well-appreciated that gauging a symmetry (i.e., non-perturbatively coupling the system to a gauge field) can significantly change the physics at play, there has been relatively little research on exactly *how* different the gauged and ungauged theories are. For instance, what is the quantum phase diagram of the model that interpolates between them?

Gauge theories are most tractable in one spatial dimension, which has been utilized by a large number of seminal works on quantum electrodynamics and non-abelian Yang-Mills theories [19–30]. These works mostly focused on the confinement[1] arising for continuous gauge groups. However, gauge theories with discrete symmetries [33–37] can exhibit deconfined phases in one dimension, e.g., in odd $\mathbb{Z}_2$ gauge theory or $\mathbb{Z}_2$ gauge theory coupled to gapless matter [38–41]. In this work, we will continue in this vein, gauging one of the most basic symmetries in Nature: $\mathbb{Z}_2^f$ fermion parity symmetry, here in the context of one-dimensional quantum lattice models.

In one spatial dimension, there are two possible gapped phases of matter in the presence of (only) fermion parity symmetry: the trivial phase and the celebrated topological phase

---

[1]Deconfinement arises at particular values of the $\theta$-angle. It has been conjectured that in these regimes, the effective low-energy description is that of a discrete gauge theory [31,32].

with zero-energy Majorana edge modes [42]. The Kitaev chain exhibits both of these phases, separated by a critical point with central charge $c = \frac{1}{2}$. In this work, we study the effect of gauging the fermion parity symmetry of this model. We approach this in two complementary ways:

- We first gauge the Kitaev chain in the traditional sense: using the framework of lattice gauge theory, one enlarges the Hilbert space by including link variables (representing the gauge field) and one subsequently again shrinks the Hilbert space by imposing a local gauge constraint that locks matter and gauge field together with a Gauss law. This is an invasive procedure that radically alters the original phase of matter.

  Doing so, we find a few novel features which had not been pointed out in previous works [43–46]. In particular, after gauging, the trivial phase of the Kitaev chain becomes a deconfined phase which for a positive chemical potential is stable to vortices, despite matter being massive, due to spontaneous translation symmetry-breaking giving rise to deconfined domain walls. Secondly, the topological Kitaev chain becomes a Higgs phase with non-trivial symmetry-protected topological (SPT) order [47–51] in the absence of vortices. More generally, we uncover that whilst gauging fermion parity eliminates all fermionic excitations in the bulk, the Hilbert space still has local, gauge-invariant fermionic operators near the boundaries of the system. In a sense, near the boundary, the global fermion parity symmetry survives gauging. This has been observed before in Ref. [52] where this duality between a global and boundary symmetry was interpreted as a holographic duality, although it was not realized that the resulting phase is a non-trivial SPT phase with topologically-protected qubits at the edge.

- An alternative way of gauging the Kitaev chain that we undertake in this paper is the following: we start from the ordinary Kitaev model and smoothly deform it, such that all quantum states that are not invariant under gauge transformations get an energetic penalty. If the penalty is large compared to energy scales of the Kitaev chain, at low-energies one is left with the gauged Kitaev model. In other words, the Gauss law is not a hard constraint in this approach, but is implemented energetically. This method allows us to interpolate gradually between the ordinary and gauged theories within a unified framework and henceforth will be called *gentle gauging*[2] in this paper. We undertake a comprehensive investigation of the quantum phase diagram of the gently gauged Kitaev model. We discover four distinct phases and investigate the nature of the quantum phase transitions separating them. Such quantum phase diagrams can be probed in future quantum simulators, where violation of the strict Gauss law in $\mathbb{Z}_2$ gauge theories [54–58] can be controlled at will using Floquet engineering [59–61]. Furthermore, using gentle gauging we show that in the absence of vortices the fermionic SPT order of the Higgs phase is in the universality class of a stacked pair of Kitaev chains.

The remainder of the paper is structured as follows: In section 2 we provide a streamlined introduction to the Kitaev model, emphasizing aspects that are relevant for our forthcoming investigation. Next, in section 3 we gauge the Kitaev chain using the Hamiltonian formalism and discuss in some detail salient features of its quantum phase diagram. Section 4 is dedicated to the gently gauged Kitaev model and its phase diagram, which sheds new light on both the ordinary and the gauged Kitaev chain. In section 5 we explore what happens in the absence of superconductivity; we find that this leads to an intrinsically gapless SPT phase [62]. Numerical methods are discussed in Section 6 and our conclusions and outlook on future research are summarized in section 7. In appendix A we provide an alternative derivation of

---

[2]One may also call it *soft gauging*, but that term is sometimes used for the special case where the Hamiltonian commutes with the Gauss operator [53].

the Hamiltonian of the gauged Kitaev chain in the spin formulation. Finally, in appendix B we gauge the one-dimensional transverse Ising model (TFIM) and discuss in what sense its phase diagram differs from the phase diagram of the gently gauged Kitaev chain.

## 2 The Kitaev chain

The Kitaev chain is a tight-binding model of spinless fermions living on the sites of a lattice with nearest neighbor hopping and pairing, described by the Hamiltonian [42, 63]

$$
H = -t \sum_j \left( c_j^\dagger - c_j \right)\left( c_{j+1}^\dagger + c_{j+1} \right) - \mu \sum_j \left( c_j^\dagger c_j - \frac{1}{2} \right) \tag{1}
$$
$$
= i t \sum_j \tilde{\gamma}_j \gamma_{j+1} + \frac{i\,\mu}{2} \sum_j \tilde{\gamma}_j \gamma_j \,.
$$

Here we have used the convenient Majorana operators

$$
\gamma_i = c_i^\dagger + c_i \qquad \text{and} \qquad \tilde{\gamma}_i = i\left( c_i^\dagger - c_i \right), \tag{2}
$$

which satisfy the hermiticity conditions $\gamma_i^\dagger = \gamma_i$ and $\tilde{\gamma}_i^\dagger = \tilde{\gamma}_i$, and the anticommutation relations $\{\gamma_i, \gamma_j\} = 2\delta_{ij}$, $\{\tilde{\gamma}_i, \tilde{\gamma}_j\} = 2\delta_{ij}$ and $\{\gamma_i, \tilde{\gamma}_j\} = 0$.

### 2.1 Topological order

This simple Hamiltonian has been studied extensively, and is the paradigmatic example of a system exhibiting two quantum phases which are not distinguishable by a local order parameter. For $\left|\frac{t}{\mu}\right| > \frac{1}{2}$ (weak pairing), the chain is in the topological phase, characterized by the presence of robust Majorana edge modes which are protected by the $\mathbb{Z}_2^f$ fermionic parity symmetry. These edge modes can be constructed exactly for a half-infinite chain: if we define

$$
\gamma_l := \sum_{j=1}^\infty \left( -\frac{\mu}{2t} \right)^{j-1} \gamma_j \,, \tag{3}
$$

then it is straightforward to show that $[\gamma_l, H] = 0$. Hence, if $|\mathrm{gs}\rangle$ is a ground state, then so is $\gamma_l |\mathrm{gs}\rangle$. Since these two states have opposite fermion parity $P = (-1)^{\sum_j n_j}$, they cannot be linearly dependent. The ground state is thus twofold degenerate with boundaries, whereas it can be shown to be unique in the absence of boundaries. Note that the edge mode in Eq. (3) is only localized for $\left|\frac{t}{\mu}\right| > \frac{1}{2}$. Indeed, for $\left|\frac{t}{\mu}\right| < \frac{1}{2}$ (strong pairing), the phase is trivial: it does not exhibit edge modes and has a unique ground state (independent of boundary conditions). These two phases cannot be connected whilst preserving an energy gap, which indeed vanishes for $\left|\frac{t}{\mu}\right| = \frac{1}{2}$ [42].

Whilst such topological order cannot be probed by local order parameters, it can be identified with nonlocal ones. If we define the semi-infinite string operators

$$
\mathcal{S}_j^{\mathrm{triv}} = (-1)^{\cdots + n_{j-2} + n_{j-1}} = \prod_{k<j} (i\tilde{\gamma}_k \gamma_k), \tag{4}
$$

$$
\mathcal{S}_j^{\mathrm{top}} = (-1)^{\cdots + n_{j-2} + n_{j-1}} (c_j^\dagger + c_j) = \left( \prod_{k<j} (i\tilde{\gamma}_k \gamma_k) \right) \gamma_j, \tag{5}
$$

then it can be shown that the trivial phase has long-range order in $\lim_{|i-j|\to\infty}\left|\langle\mathcal{S}_i^{\text{triv}}\mathcal{S}_j^{\text{triv}}\rangle\right|\neq 0$, whereas the topological phase has long-range order in $\lim_{|i-j|\to\infty}\left|\langle\mathcal{S}_i^{\text{top}}\mathcal{S}_j^{\text{top}}\rangle\right|\neq 0$. The discrete invariant distinguishing these two cases is the charge of the string order parameter under fermion parity: $P\mathcal{S}^{\text{triv}}P=\mathcal{S}^{\text{triv}}$ whereas $P\mathcal{S}^{\text{top}}P=-\mathcal{S}^{\text{top}}$. Indeed, it can be argued that having long-range order in a string order parameter that is odd under $P$ is sufficient to deduce the existence of zero-energy Majorana modes in the presence of boundaries.

We note that it is sometimes said that instead of having strict topological order, the Kitaev chain is a symmetry-protected topological (SPT) phase, in this case protected by the fermion parity symmetry $P$. The reason for saying this is because it naturally fits into the general SPT framework (as is also evidenced by the fact that the bulk order parameter has a string consisting of the protecting symmetry, which is a common theme for SPT phases). However, it is important to keep in mind that it is impossible to break fermion parity symmetry whilst preserving locality; it is thus an automatic symmetry of any fermionic system.

## 2.2 Jordan-Wigner and the transverse-field Ising model

Before gauging the model, there are good reasons to briefly recap the Jordan-Wigner (JW) mapping of the Kitaev model to a spin chain. Firstly, we find (in section 3) that gauging the Kitaev chain also gives rise to a spin chain which is related, but distinct from the JW spin chain. Secondly, the Jordan-Wigner transformation and gauging can both be seen as two distinct types of bosonization. We will explain the interrelation between these concepts in detail in section 3.

Long before the topological properties of Hamiltonian (1) were fully appreciated by Kitaev, it was known as the JW dual of the transverse-field Ising chain (TFIM) [64], given by the following spin-1/2 Hamiltonian:

$$H_{\text{TFIM}}=-t\sum_i\tau_i^x\tau_{i+1}^x+\frac{\mu}{2}\sum_i\tau_i^z.\tag{6}$$

Here the JW transformation is defined by

$$\tau_j^x=(-1)^{\sum_{k<j}n_k}\gamma_j\qquad\text{and}\qquad\tau_j^z=2n_j-1=i\tilde{\gamma}_j\gamma_j,\tag{7}$$

where $n_j=c_j^\dagger c_j$ denotes the number operator, which indeed maps Eq. (1) to Eq. (6). Since this transformation is non-local, it can drastically alter the physics of the system. In this case, we see that it maps the topological phase to a symmetry-breaking phase, with the topological string order parameter $\mathcal{S}^{\text{top}}$ becoming the local Ising order parameter. The JW transformation is a unitary map for open boundaries, and indeed, for this geometry both the topological Kitaev chain and the Ising phase have a twofold ground state degeneracy. However, for periodic boundary conditions, using Eq. (7) would generate an additional non-local term, which looks unnatural in the spin chain language. In absence of this unnatural term, $H$ in Eq. (1) and $H_{\text{TFIM}}$ are not unitarily equivalent, the former having a unique ground state (in the topological phase) whereas the latter is still twofold degenerate due to symmetry-breaking.

We will now turn to gauging, and we will see that—similar but distinct to the above JW transformation—it drastically changes the physics of the original model.

# 3 Gauging the Kitaev chain

Any closed fermionic system has the fermionic parity symmetry $P=(-1)^{\sum_j n_j}$. As we saw in section 2, there are two one-dimensional fermionic phases of matter in this symmetry class. In this section, we will be gauging this symmetry and exploring what happens to these two phases. Section 4 will explore the relationship between the gauged and ungauged models.

Lattice gauge theory was introduced by Wegner in 1971 [33] for discrete groups and by Wilson in 1974 [65] for continuous groups[3]. The path integral approach of these seminal works was extended by Kogut and Susskind to the Hamiltonian formalism [67], culminating in the famous review by Kogut [68]. In this paper we will follow the latter approach which will be presented in a self-contained manner.

## 3.1 The Hilbert space

Let us briefly recap how to put electrodynamics—where the gauge group is $U(1)$—on a lattice, which will motivate the notion of a discrete lattice gauge theory. Starting with the global $U(1)$ symmetry of a fermion $\psi(x) \to e^{i\lambda}\psi(x)$, we promote it to a local symmetry $\psi(x) \to e^{i\lambda(x)}\psi(x)$ at the cost of introducing a new field which transforms as $A(x) \to A(x) + \partial_x \lambda(x)$. If we discretize this in a one-dimensional geometry, it is natural to put $\psi$ on sites and $A$ on links such that the derivative $\partial_x \lambda$ can be approximated by a finite difference:

$$\psi_j \to e^{i\lambda_j}\psi_j \qquad \text{and} \qquad A_{j+\frac{1}{2}} \to A_{j+\frac{1}{2}} + \lambda_{j+1} - \lambda_j, \tag{8}$$

where $j + 1/2$ labels the link between sites $j$ and $j + 1$ (with $j$ integer).

To consider a $\mathbb{Z}_2$ gauge theory, we can restrict the global charge symmetry to its parity subgroup, i.e., we restrict $\lambda_j$ and $A_{j+1/2}$ to take values in $\{0, \pi\}$. If we introduce the notation $c_j := \psi_j$ and $\sigma^z_{j+1/2} := e^{iA_{j+1/2}}$, then a gauge transformation is just given by the sign $s_j := e^{i\lambda_j} \in \{-1, 1\}$ such that

$$c_j \to s_j c_j \qquad \text{and} \qquad \sigma^z_{j+\frac{1}{2}} \to s_j \sigma^z_{j+\frac{1}{2}} s_{j+1}. \tag{9}$$

We see that this is generated[4] by the Gauss operator

$$G_j := \sigma^x_{j-\frac{1}{2}} (-1)^{n_j} \sigma^x_{j+\frac{1}{2}}. \tag{10}$$

Since we defined $\sigma^z_{j+1/2} = e^{iA_{j+1/2}}$, we can naturally associate[5] $\sigma^x_{j+1/2} = e^{iE_{j+1/2}}$, i.e., it is the exponential of the electric field. Hence, the condition of gauge-invariance—namely that every quantum state is invariant under $G_j$—can be interpreted as saying that the divergence of the electric field around a given site is given by whether or not that site is occupied, mimicking the Gauss law $\nabla \cdot E = \rho$.

In conclusion, the naive Hilbert space consists of site variables—which are fermionic—and link variables—which are spin-1/2 degrees of freedom. Crucially, gauging imposes a local constraint and we will only consider states which are invariant[6] under $G_j$

$$\mathcal{H}_{\text{phys}} = \left\{ |\psi\rangle \in \mathcal{H}_{\text{sites}} \otimes \mathcal{H}_{\text{links}} \mid \forall j : G_j|\psi\rangle = |\psi\rangle \right\}. \tag{11}$$

We sketch this in Fig. 1, where filling $n_j = c^\dagger_j c_j = 0, 1$ is shown by white and black dots, and whether or not there is an electric field $\sigma^x_{j+1/2}$ on a given link is denoted by a solid or dashed line. We explicitly show the combinations allowed by the Gauss law $G_j = +1$.

The above is clear-cut when we are in the bulk of the system. We have not yet fully specified the problem near the boundary, where the Gauss operator might not even be well-defined. We postpone this discussion to section 3.4.

---

[3]A finite-dimensional version for continuous groups was later introduced by Chandrasekharan and Wiese, known as quantum link models [66].

[4]Conjugating any operator by $G_{j_0}$ implements the gauge transformation for $s_j = 1 - 2\delta_{j,j_0}$.

[5]Note that this is equivalent to $A_{j+1/2} = \frac{\pi}{2}\left(1 - \sigma^z_{j+1/2}\right)$ and $E_{j+1/2} = \frac{\pi}{2}\left(1 - \sigma^x_{j+1/2}\right)$. Since the local Hilbert space is finite-dimensional, $E$ and $A$ do not satisfy the same commutation relations as we are used to in quantum electrodynamic [69].

[6]We thus gauge the fermion parity symmetry in the absence of static charges.

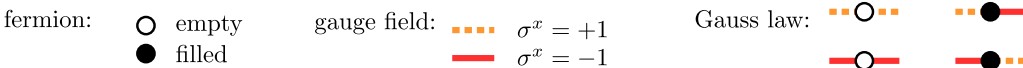

Figure 1: The Hilbert space of the $\mathbb{Z}_2$ gauge theory, which can be made from fermionic sites and spin-1/2 links. The only allowed combinations are those satisfying the Gauss law.

## 3.2   The Hamiltonian

Having established the Hilbert space, we now embed the Kitaev chain (1) into it by coupling it to a $\mathbb{Z}_2$ gauge field. The general procedure for making a Hamiltonian gauge-invariant is called *minimal coupling*. Again, we first remind the reader of what this looks like in the continuum, before showing the analogous lattice set-up. It comes down to replacing the gauge-dependent operator $\psi(x)$ by the gauge-invariant[7] $e^{-i\int_{-\infty}^{x} A(x')\mathrm{d}x'}\psi(x)$. Note that this substitution rule indeed maps the kinetic term $\psi(x)^{\dagger}\partial_x \psi(x)$ to the familiar $\psi(x)^{\dagger}(\partial_x - iA(x))\psi(x)$. The analogue of minimal coupling for $\mathbb{Z}_2$ lattice gauge theory is given by the substitution

$$c_j \quad \rightarrow \quad \cdots \sigma_{j-\frac{3}{2}}^z \sigma_{j-\frac{1}{2}}^z c_j = \Big(\prod_{k \leq j} \sigma_{k-\frac{1}{2}}^z\Big)c_j \,. \tag{12}$$

This shows the non-local and invasive nature of gauging, and is reminiscent of bosonization or the Jordan-Wigner transformation. We will come back to this similarity in section 3.8. This substitution rule maps the kinetic term in Eq. (1) as follows:

$$it \sum_j \tilde{\gamma}_j \gamma_{j+1} \quad \rightarrow \quad it \sum_j \tilde{\gamma}_j \sigma_{j+\frac{1}{2}}^z \gamma_{j+1} \,, \tag{13}$$

where we remind the reader that $\gamma_j = c_j^{\dagger} + c_j$ and $\tilde{\gamma}_j = i\left(c_j^{\dagger} - c_j\right)$. Note that this substitution does not affect the chemical potential term.

In addition to making the fermionic Hamiltonian gauge-invariant, we can add kinetic and potential terms for the gauge field. The customary way of doing that in the Lagrangian/space-time picture is by adding local Wilson loops $e^{i\oint A}$ to the action. Indeed, in the continuum limit of the lattice model for a continuous gauge group, an elementary Wilson loop reproduces the Yang-Mills action[8] [65]. Given that our model lives in only one spatial dimension, there are no (local) Wilson loops that lie entirely within the same time-slice—there are no magnetic fields in one spatial dimension. However, if for the moment we also think of time being discrete, then we can form a Wilson loop $\prod_l \sigma_l^z$ around a plaquette in spacetime. When going from the Lagrangian picture to a Hamiltonian picture, we have to (partially) fix our gauge in the temporal direction, and doing so[9] changes this spacetime Wilson loop into an electric field operator $\sigma_{j+1/2}^x$ [68].

---

[7]In fact, the phase picks up a factor at negative infinity, but it will ultimately drop out for any Hamiltonian which is invariant under the symmetry that is gauged.

[8]More precisely, in Euclidean space, the Wilson loop in the $(\mu, \nu)$ plane will give $e^{i\oint A} \approx 1 + ia^2 F_{\mu\nu} - \frac{1}{2}a^4(F_{\mu\nu})^2 + \cdots$, such that summing over all planes gives the Yang-Mills Lagrangian as the leading contribution.

[9]After we use the gauge symmetry to fix the spins on the temporal links to point up, our Wilson loop looks like an Ising coupling $\sigma^z \sigma^z$ connecting two different time-slices. By the usual classical-quantum corresponendence, this becomes a transverse field $\sigma^x$ in the Hamiltonian.

In summary, the gauged Kitaev chain is given by

$$H = -t \sum_j \left( c_j^\dagger - c_j \right) \sigma_{j+1/2}^z \left( c_{j+1}^\dagger + c_{j+1} \right) - \mu \sum_j \left( c_j^\dagger c_j - \frac{1}{2} \right) - h \sum_j \sigma_{j+1/2}^x \tag{14}$$

$$= i\, t \sum_j \tilde{\gamma}_j \sigma_{j+1/2}^z \gamma_{j+1} + \frac{i\,\mu}{2} \sum_j \tilde{\gamma}_j \gamma_j - h \sum_j \sigma_{j+1/2}^x,$$

with the gauge constraint/Gauss law

$$G_j = \sigma_{j-1/2}^x (-1)^{n_j} \sigma_{j+1/2}^x = \sigma_{j-1/2}^x\, i\tilde{\gamma}_j \gamma_j\, \sigma_{j+1/2}^x = +1. \tag{15}$$

Changing the sign of $t$ leads to a unitarily equivalent model, obtained through the transformation $\sigma^z \to -\sigma^z$, $\sigma^y \to -\sigma^y$, $\sigma^x \to \sigma^x$. A similar consideration holds for $h \to -h$, and therefore in the following we will only consider $h, t \geq 0$. If $h = 0$, then the sign of $\mu$ can also be toggled by a unitary transformation. However, for $h \neq 0$, the sign will be important and we will discuss both cases separately. We stress that although the $h$-term in Eq. (14) was motivated by fixing a gauge on temporal links, the term itself is manifestly gauge-invariant in this quantum lattice gauge theory—indeed, it commutes with the gauge constraint in Eq. (15). Moreover, let us mention that although the case $h = 0$ can mapped to a free-fermion problem, this requires a nonlocal mapping which can obscure some of the relevant physics, so we do not follow this route.

Note that according to [70], the gauged Kitaev model (14) (yet without the Gauss constraint (15)) can be realized in a helical quantum wire proximity coupled to a superconductor with quantum phase slips. That model exhibits an interesting relation between thermal conductance and confinement [70].

## 3.3 Symmetries

In addition to the local (gauge) symmetry, the gauged Kitaev model (14) has a physical global symmetry $\prod_j \sigma_{j+1/2}^z$ if $h = 0$. One can interpret this as a Wilson loop around the system, $e^{i \int A}$. Indeed, even though there is no local magnetic field in one spatial dimension, one can still measure whether or not there is a flux piercing the total system when placed on a circle. For this reason we will denote this symmetry as $W := \prod_j \sigma_{j+1/2}^z$, which is sometimes also referred to as the *magnetic symmetry*. This symmetry tells us that the global magnetic flux is preserved. Operators which toggle this flux are called *vortices* (in spacetime) or sometimes also *instantons*. The instanton operator is given by $\sigma_{j+1/2}^x$ (for any $j$), which indeed anticommutes with $W$. Hence, we can interpret the electric term in Eq. (14), with strength $h$, as dynamically creating instantons in the system, destroying the magnetic symmetry. Such instantons can lead to confinement, which we will study in section 3.5.

Let us note that in addition, there is also still the global fermion parity symmetry. This is a subtle symmetry: with periodic boundary conditions, the symmetry is gone, it is a pure do-nothing gauge redundancy. This can be confirmed by noting that fermion parity is simply a product of all the local Gauss operators $P = \prod_j G_j = 1$. However, for open boundary conditions this no longer needs to hold. For instance, for open chains terminating with link variables, the fermion parity is not restricted to be even. To see this, note that from the relation

$$1 = \prod_i G_i = \sigma_{\frac{1}{2}}^x \left( \prod_{i=0}^L (-1)^{n_i} \right) \sigma_{L+\frac{1}{2}}^x, \tag{16}$$

we see that the fermion parity

$$P = \sigma_{\frac{1}{2}}^x \sigma_{L+\frac{1}{2}}^x. \tag{17}$$

This demonstrates that the global total fermionic parity symmetry survives gauging, but it only acts non-trivially near the edge of the system. This suggests that, perhaps, the system is bosonic in the bulk and fermionic near the edge, which we explore and confirm in more detail now.

### 3.4   Spins in the bulk, fermions at the edge

Here we show how the bulk of the gauged Kitaev chain is *locally* equivalent to a spin chain.[10] The physical reason for the bulk being bosonic rather than fermionic is that $c_j^\dagger$ is not gauge-invariant: as discussed in section 3.2 we need to attach a gauge string to get the gauge-invariant $\left(\prod_{k \le j} \sigma_{k-1/2}^z\right) c_j^\dagger$—which we can interpret as an *emergent fermion*. Similar to those encountered in higher-dimensional gauge theories [36, 71–76], such emergent fermions can only be created in pairs, consistent with the fact that any local operator in the bulk of this theory is bosonic. There is no *local* gauge-invariant operator that creates an odd number of fermions. This is equivalent to what we observed in section 3.3, namely that fermionic parity cannot change in the bulk.

Let us introduce new *gauge-invariant* spin-1/2 variables [40, 44, 52]

$$X_{i+1/2} = \sigma_{i+1/2}^x, \qquad Y_{i+1/2} = -i\tilde{\gamma}_i \sigma_{i+1/2}^y \gamma_{i+1} \qquad \text{and} \qquad Z_{i+1/2} = -i\tilde{\gamma}_i \sigma_{i+1/2}^z \gamma_{i+1}. \quad (18)$$

We readily confirm that these are bosonic, square to one, commute when on different links, and obey the Pauli algebra when on the same link. Crucially, these commute with the local Gauss operator $G_j$ (15), making them physical. In terms of these variables, the gauged Kitaev chain Hamiltonian (14) becomes

$$H = \frac{\mu}{2} \sum_j X_{j-1/2} X_{j+1/2} - t \sum_j Z_{j+1/2} - h \sum_j X_{j+1/2}. \quad (19)$$

Here we used the Gauss law to rewrite the chemical potential term. For an alternative derivation of this Hamiltonian that relies on two non-local transformations, see appendix A. Yet another proof of the bulk equivalence of the Hamiltonian of the gauged Kitaev model and the Hamiltonian (19) was presented in the appendix of Ref. [77].

We thus see that the system is described by an Ising model in a transverse and longitudinal field (TLFIM) with no remaining gauge constraints. Its phase diagram has been studied before and shows distinct physics depending on whether it is ferromagnetic ($\mu < 0$) [78] or antiferromagnetic ($\mu > 0$) [79]; we reproduced this with iDMRG as shown in Fig. 2. However, its reinterpretation as a phase diagram of a gauge theory is novel, and we will discuss the labeling of the phases in the subsequent sections 3.5, 3.6 and 3.7. We note that the magnetic symmetry $W = \prod_j \sigma_{j+1/2}^z$ has become $W \propto P \prod_j Z_{j+1/2}$, where $P$ is global fermionic parity. For periodic boundary conditions, we have $P = 1$, such that $W$ essentially coincides with the Ising symmetry $\prod_j Z_{j+1/2}$, which is explicitly broken for $h \ne 0$.

An important subtlety for understanding the phases of matter in Fig. 2 is the realization that the effective gauge-invariant spin chain (19) is only equivalent to the gauged Kitaev chain (14) *in the bulk of the system*. Equivalently, it captures what happens for periodic boundary conditions. However, in the presence of boundaries, our new variables in Eq. (18) might not be well-defined. For instance, let us consider a finite gauged Kitaev chain (14) with sites $j = 1, \cdots, L$ such that the system begins and ends with link variables, the leftmost and rightmost bond being labeled by $1/2$ and $L+1/2$, respectively. In this geometry, the Gauss operator

---

[10]This is to be contrasted to the Jordan-Wigner transformation encountered in section 2.2, which is a non-local transformation and thus can severely change the physics at play; this is discussed in detail in section 3.8.

(15) is well-defined for every site. However, the definition of the gauge-invariant variables (18) needs to be modified for the outer links:

$$X_{1/2} = \sigma_{1/2}^x, \qquad Y_{1/2} = \sigma_{1/2}^y \gamma_1 \qquad \text{and} \quad Z_{1/2} = \sigma_{1/2}^z \gamma_1;$$
$$X_{L+1/2} = \sigma_{L+1/2}^x, \qquad Y_{L+1/2} = \tilde{\gamma}_L \sigma_{L+1/2}^y \quad \text{and} \quad Z_{L+1/2} = \tilde{\gamma}_L \sigma_{L+1/2}^z. \tag{20}$$

These are still gauge-invariant, but note that $Y$ and $Z$ are now fermionic (i.e., they anticommute with the fermion parity $P$). Our Hilbert space is thus fermionic near the edges! The Hamiltonian for this geometry is

$$H = \frac{\mu}{2} \sum_{j=1}^{L} X_{j-1/2} X_{j+1/2} - t \sum_{j=1}^{L-1} Z_{j+1/2} - h \sum_j X_{j+1/2}. \tag{21}$$

Observe that $Z_{1/2}$ and $Z_{L+1/2}$ do not appear (and indeed they cannot, given that they are fermionic), making the model manifestly distinct from the usual Ising chain in a transverse and longitudinal field. This is the first indication that the paramagnetic phase in fact has nontrivial edge physics, which we confirm in section 3.6. With regards to the magnetic symmetry, this still equals $W \propto P \prod_j Z_j$. Combining this with Eq. (17), we have that

$$W \propto Y_{1/2} Z_{1+1/2} Z_{2+1/2} \cdots Z_{L-1/2} Y_{L+1/2}. \tag{22}$$

Indeed, for $h = 0$ this is a symmetry of Eq. (21). The fact that the endpoint operator $Y$ is fermionic (see Eq. (20)) will be key to showing in section 3.6 that the paramagnetic phase is in fact a topological phase protected by magnetic symmetry and fermion parity.

Of course, we can also do the analysis for the other geometry, where the chain ends with site variables rather than link variables. In this case, the Gauss operator $G_j$ is only well-defined for sites $j = 2, 3, \cdots, L-1$. Hence, we again end up with fermionic variables on sites 1 and $L$. One could choose to introduce new Gauss operators on these sites, of the form $G_1 = (-1)^{n_1} \sigma_{1+1/2}^x$ and $G_L = \sigma_{L-1/2}^x (-1)^{n_L}$. In this case there are *no* fermionic degrees of freedom, even at the edge. However, these new Gauss operators completely break magnetic symmetry. We can thus summarize as follows: whilst the bulk of the gauged Kitaev chain is a purely bosonic system, the edge is fermionic as long as the Gauss law respects magnetic symmetry.

## 3.5 The deconfined and Higgs phases

We now analyze the ground state phase diagram of Eq. (19) as plotted in Fig. 2. In this section and the next, we set $h = 0$ (see section 3.7 for $h \neq 0$), in which case the model (19) simplifies to

$$H = \frac{\mu}{2} \sum_j X_{j-1/2} X_{j+1/2} - t \sum_j Z_{j+1/2}. \tag{23}$$

In the regime $0 < t \ll |\mu|$, the dominant term in the Hamiltonian is the Ising coupling $\sum_j X_{j-1/2} X_{j+1/2}$, such that the ground state spontaneously breaks the magnetic/Ising symmetry $W$. For $\mu > 0$, the ground state moreover spontaneously breaks translation symmetry, whereas for $\mu < 0$ the ground state is ferromagnetic (this will be important when we turn on $h \neq 0$ in section 3.7). It is well-known that in one spatial dimension, domain wall excitations in symmetry-breaking phases are deconfined. Such domain walls are created by the semi-infinite string operator

$$D_j = \prod_{k<j} Z_{k+1/2}. \tag{24}$$

A local operator can only create pairs of domain walls, such as $D_j D_{j+1}$. However, these are dynamically deconfined and will spread out indefinitely with no string tension. Indeed, if

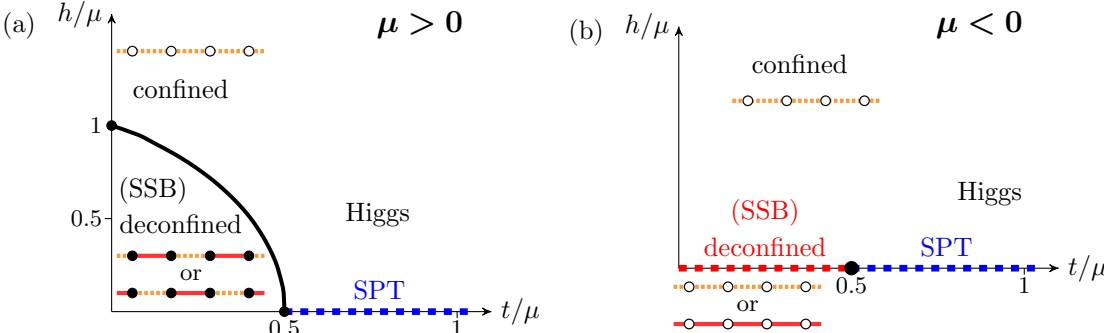

Figure 2: Phase diagram of the gauged Kitaev chain (14) for $\mu > 0$ and $\mu < 0$, respectively. In absence of vortices ($h = 0$), the system enjoys magnetic symmetry, protecting an SPT order in the Higgs phase (highlighted by dashed blue line). For $h \neq 0$, the Higgs and confined regimes are adiabatically connected. For $\mu > 0$ the solid black line denotes Ising criticality.

there would be any tension, then that would mean that energetically one of the two ground states is preferred over the other, whereas spontaneous symmetry breaking implies an exact degeneracy between the two ground states (in the thermodynamic limit). To relate this to the gauge theory, we use Eq. (18) to rewrite

$$D_j = \left( \prod_{k<j} (-1)^{n_k} \sigma^z_{k+1/2} \right) \gamma_j = \left( \prod_{k<j} \sigma^z_{k+1/2} \right) \sigma^x_{j-1/2} \gamma_j \,, \tag{25}$$

where in the last step we used the Gauss law. The main point is that the domain wall operator $D_j$ is fermionic and gauge-invariant. Hence, the deconfinement of domain walls exactly coincides with the deconfinement of the (emergent) fermionic matter in the $\mathbb{Z}_2$ gauge theory.

Due to the well-known exact solubility of Eq. (23), we know that this deconfined phase persists for $t < |\mu|/2$. For $t > |\mu|/2$—which is the gauged version of the topological Kitaev chain phase—we enter the paramagnetic phase, which we claim to be the Higgs phase of the gauge theory. The Higgs phase is defined by charges being condensed into the ground state. In other words, we require that there is long-range order $\lim_{|i-j| \to \infty} \langle D_i D_j \rangle \neq 0$ for $D_j$ in Eq. (25). But this exactly coincides with the known long-range order of the domain wall operator in the paramagnetic phase. Indeed, this is the Kramers-Wannier dual of the long-range order of $X_j$ in the symmetry-broken phase.

In the fixed point limit where $t \to +\infty$, we see that the ground state is given by $Z_{j+1/2} = 1$. This naively looks like a product state, but we have to remember that $Z_{j+1/2}$ is a composite object defined in Eq. (18). In fact, the ground state has non-trivial entanglement and we will now explain that the Higgs phase forms a topologically non-trivial phase of matter. Similary, the Ising transition separating the deconfined and Higgs phase is also topologically non-trivial, being an instance of non-trivial symmetry-enriched quantum criticality [80].

## 3.6 Higgs = SPT

As indicated by the dashed line in Fig. 2, we claim that for $h = 0$ the Higgs phase is in a fermionic[11] symmetry-protected topological (SPT) phase, protected by the magnetic symmetry $W$ and the fermion parity symmetry $P$.

Let us first analyze this at the boundary of the system, where we will find a two-dimensional zero-energy mode at each edge. As discussed in section 3.4, for a system with boundaries,

---

[11]An SPT phase is called fermionic when it cannot exist in a purely-bosonic Hilbert space.

beginning and ending with link variables, we have

$$H = \frac{\mu}{2} \sum_{j=1}^{L} X_{j-1/2} X_{j+1/2} - t \sum_{j=1}^{L-1} Z_{j+1/2} \,, \tag{26}$$

where we have set $h = 0$ in Eq. (21), with $X, Y, Z$ defined in Eqs. (18) and (20). Remember that fermion parity symmetry prevents us from adding a transverse field to the leftmost and rightmost links, i.e., $Z_{1/2}$ and $Z_{L+1/2}$, since they are fermionic as defined in Eq. (20). This implies that $X_{1/2}$ and $X_{L+1/2}$ commute with the Hamiltonian: in other words, we can think of them as symmetries. Moreover, they anticommute with the global Ising symmetry. Having two anticommuting symmetries (say, $X_{1/2}$ and $W$) already tells us that the ground state[12] will be twofold degenerate.[13] In the deconfined phase, this is simply restating the bulk degeneracy due to spontaneous symmetry breaking. However, in the Higgs phase, we saw that with periodic boundary conditions, the Hamiltonian is equivalent to the usual paramagnetic phase, which has a unique ground state. Hence, in the Higgs phase, this degeneracy is associated to having an edge. In fact, there is a second commuting edge operator defined by

$$\gamma_l = Y_{1/2} - \frac{\mu}{2t} Z_{1/2} Y_{1+1/2} + \left(-\frac{\mu}{2t}\right)^2 Z_{1/2} Z_{1+1/2} Y_{2+1/2} + \cdots. \tag{27}$$

A straightforward computation shows that $[\gamma_l, H] = O\left(\left(-\frac{\mu}{2t}\right)^L\right)$, i.e., $\gamma_l$ is an exponentially-localized zero-energy mode (with an exponentially-small finite-size energy splitting) in the Higgs phase ($|\mu| < 2t$). Moreover, remembering the definition (20), we see that $\gamma_l$ is fermionic (indeed, it is a normalizable Majorana mode).

In conclusion, in the Higgs phase at $h = 0$, the left edge has two localized edge mode operators that commute with the Hamiltonian but anticommute with one another, $\gamma_l$ and $X_{1/2}$, giving us a localized twofold ground state degeneracy at the left edge. Fermion parity $P$ prevents us from adding $\gamma_l$ to the Hamiltonian, and magnetic symmetry $W$ prevents us from adding $X_{1/2}$; the edge qubit is thus protected! Arguing similarly at the right edge, we conclude that the open chain has four-fold ground state degeneracy with an exponentially-small finite-size energy splitting. To be more precise, the four-dimensional ground state manifold is formed by a pair of two strictly degenerate eigenstates which are separated by an exponentially-small energy gap. The same result is obtained by repeating this analysis for boundaries ending with sites rather than link variables—on the condition that the Gauss law preserves magnetic symmetry.

We demonstrated the existence and stability of the edge mode constructively. But the reader might wonder why they are there in the first place. In the above discussion, they appeared as if by magic. However, we can interpret them as naturally arising from the notion of symmetry fractionalization, which explains all SPT phases (for a review, see e.g. Ref. [82]). More precisely, we can interpret the two edge mode operators $X_{1/2}$ and $\gamma_l$ as encoding the effective symmetry action of $P$ and $W$ on the boundary, respectively. For convenience, let us work in the limit $\mu \to 0$, such that the edge mode operators are $X_{1/2}$ and $\gamma_l = Y_{1/2}$, see Eq. (27). Remember that for a system with boundaries, the fermion parity could be written as

$$P = \sigma_{1/2}^x \sigma_{L+1/2}^x = X_{1/2} X_{L+1/2} \,, \tag{28}$$

---

[12]In fact, for this fine-tuned Hamiltonian, we see that the degeneracy applies to the whole spectrum; we say this is a *strong* edge mode in the sense of Ref. [81].

[13]Let $|\psi\rangle$ be a ground state. If either $X_{1/2}|\psi\rangle$ or $W|\psi\rangle$ are linearly independent from $|\psi\rangle$, we are done. Otherwise, write $X_{1/2}|\psi\rangle = \lambda|\psi\rangle$ and $W|\psi\rangle = \lambda'|\psi\rangle$ (with $\lambda, \lambda' \in U(1)$). Then $\lambda\lambda'|\psi\rangle = X_{1/2}W|\psi\rangle = -WX_{1/2}|\psi\rangle = -\lambda'\lambda|\psi\rangle$, which is in clear contradiction with the fact that $\lambda$ and $\lambda'$ are commuting numbers.

as derived in section 3.3. Since $P$ must clearly commute with the Hamiltonian, and since the Hamiltonian is local, this tells us that $X_{1/2}$ is a local integral of motion. Similarly, in section 3.4 we derived that

$$W \propto -Y_{1/2}Z_{1+1/2}Z_{2+1/2}\cdots Z_{L-1/2}Y_{L+1/2}. \tag{29}$$

In the fixed point limit where $\mu = h = 0$, we have that $Z_{j+1/2} = 1$ (except for the boundary links), such that effectively

$$W \propto Y_{1/2}Y_{L+1/2}. \tag{30}$$

Since $W$ is a symmetry and since the Hamiltonian is local, we can again conclude that $Y_{1/2}$ is a local integral of motion. This way, we have derived our two edge mode operators from symmetry principles. In the latter derivation (for $W$), we made our lives simple by working in the fixed-point limit of the Higgs phase. However, the idea that one can effectively write $W \approx W_l W_r$ (where $W_l$ and $W_r$ only act near the boundary with an exponentially small tail into the bulk) is applicable to any gapped phase of matter that does not break the symmetry. This can either be derived using the matrix product state formalism, or more physically using the idea that $W \approx 1$ for periodic boundary conditions and the fact that the state has a finite correlation length (for a more detailed discussion, see Ref. [82]). In the fixed-point limit, we were able to explicitly derive that $W_l = Y_{1/2}$. From this, we infer the important property that $PW_lP = -W_l$, i.e., the magnetic and fermion parity $\mathbb{Z}_2$ symmetries are realized projectively on the edge. This discrete property of $W_l$ cannot change as long as it is well-defined, i.e., as long as the system remains gapped and symmetric. This gives us a discrete SPT invariant, putting the system in the same phase of matter as a stack of *two* Kitaev chains, protected by the fermion parity of a *single* chain. In fact, in section 4 this relationship will become very apparent.

The fact that the Higgs phase is a non-trivial SPT phase can also be detected in the bulk, e.g., by using string order parameters. This perspective shows that it is in fact inevitable: from concatenating Gauss laws, we see that the ground state has long-range order in

$$\left\langle \sigma^x_{i-1/2}(-1)^{\sum_{i \le k \le j} n_k}\sigma^x_{j+1/2} \right\rangle = 1. \tag{31}$$

This can be interpreted as a string order parameter for the fermion parity symmetry, whose endpoint operator is odd under the magnetic symmetry. Since this is an automatic consequence of the Gauss law, we see that *any magnetic-symmetry-preserving phase in the gauge theory must be a non-trivial SPT phase!* This more general perspective is worked out in greater detail (e.g., in higher dimensions) in an upcoming work [83]. Equivalently, we can look at the string order parameter associated to the magnetic symmetry $W$. This is in fact given by the domain wall operator (25), and again, we see that its endpoint operator is charged under $P$, signifying a non-trivial topological phase of matter. Given that this is unavoidable (indeed, we cannot realize the trivial phase in our gauge theory), one might wonder whether it remains meaningful to think of it as non-trivial[14]. The fact that it has protected edge modes is the most clear-cut way of seeing that this is indeed meaningful. In fact, one can think of the 'vacuum' on the outside of the system as being a truly trivial phase, as distinct from the Higgs phase. We will be able to make this point more explicit using gentle gauging in section 4.

Before addressing the effects of turning on $h \ne 0$, let us note that while the $\gamma_l$ edge mode operator (27) delocalizes as we approach the Ising criticality to the deconfined phase, the other edge mode operator, $X_{1/2}$, remains. This means that the critical system with open boundaries exhibits exact twofold degeneracy of the energy spectrum, whereas this does not occur for periodic boundary conditions. (If one tunes beyond the critical point, this becomes the twofold

---

[14]The authors are reminded of the zen koan about a tree falling in the woods.

symmetry-breaking degeneracy.) In particular, the critical ground state thus forms a topologically non-trivial gapless phase in the sense of Refs. [80,84]—where the Ising criticality for the $\mathbb{Z}_2$ magnetic symmetry $W$ is enriched by the fermionic parity symmetry $P$.

## 3.7 Vortices and confinement

When discussing the deconfined and Higgs phases above, we focused so far on the case $h = 0$, corresponding to the horizontal axis in Fig. 2. We now consider $h \neq 0$. As discussed in section 3.3, this introduces vortices (or instantons) into the system, explicitly breaking the magnetic $\mathbb{Z}_2$ symmetry $W$. Its effect on the Higgs phase is immediate: the protected edge mode is lifted, and as shown in Fig. 2, the phase is adiabatically connected to the limit $h \to +\infty$, where the ground state is given by the product state $X_{j+1/2} = \sigma^x_{j+1/2} = 1$.

The effect of adding vortices is more interesting for the deconfined phase. Since we *explicitly* break the magnetic symmetry—whose *spontaneous* breaking was key to having deconfined charges—one might expect that this necessarily leads to confinement. Indeed, this happens for $\mu < 0$, as shown in Fig. 2(b). However, for $\mu > 0$, the deconfined phase also spontaneously breaks translation symmetry, since the Ising term in Eq. (19) is antiferromagnetic. In particular, it breaks single-site translation symmetry ($\cong \mathbb{Z}$) down to its two-site translation subgroup ($\cong 2\mathbb{Z}$), with the symmetry-broken ground state manifold described by the quotient $\mathbb{Z}/(2\mathbb{Z}) \cong \mathbb{Z}_2$. Since the resulting domain walls are still fermionic, deconfinement is stable, and it can only be undone by eventually encountering an Ising transition which restores translation symmetry, as shown in Fig. 2(a). A similar discussion can be found in Ref. [38], which studies a pure Ising gauge theory. In our model this is obtained in the limits $\mu \to \pm\infty$, where the Gauss law becomes $X_{j-1/2}X_{j+1/2} = \mp 1$, referred to as odd or even Ising gauge theory, respectively. From the above discussion, we learn that even (odd) Ising gauge theory is confined (deconfined) in one spatial dimension.

Another way of understanding why the deconfined phase at $\mu > 0$ is stable to $h \neq 0$ is the realization that the true instanton operator is $(-1)^j \sigma^x_{j+1/2}$, which would indeed immediately lead to confinement. Since our Hamiltonian has translation symmetry, this operator cannot be generated. In other words, the presence of additional crystalline symmetries prevents us from the usual confinement mechanism. This is similar to what was studied by Lai and Motrunich in a spin liquid ladder [39]; more generally, having monopoles which carry non-trivial charge under crystalline symmetries is also key to many known instances of deconfined quantum criticality [85].

To get further insight into the competition between the deconfined and confined phases, it is interesting to set $t = 0$, corresponding to the vertical axis in Fig. 2(a), where the effective spin model (19) becomes

$$H = \frac{\mu}{2} \sum_j X_{j-1/2}X_{j+1/2} - h \sum_j X_{j+1/2}. \tag{32}$$

This can be seen as a classical model, since it is diagonal in the eigenbasis of $X_{j+1/2}$. It is useful to rewrite this (up to a global constant) as

$$H = 2\mu \sum_j \mathcal{P}_{j-1/2}\mathcal{P}_{j+1/2} + (\mu - h) \sum_j X_{j+1/2} \quad \text{with } \mathcal{P}_{j-1/2} := \frac{1 - X_{j-1/2}}{2}. \tag{33}$$

The operator $\mathcal{P}_{j-1/2}$ is a projector onto a down spin in the $X$ basis. Hence, if $\mu = h$ (such that the second term disappears), the first term energetically punishes all states where two neighboring spins point down. These degenerate ground states span a Hilbert space without a tensor product structure, with a number of states asymptotically given by $\phi^N$, where $\phi = (1 + \sqrt{5})/2$

Figure 3: Relations between the gauged and ungauged Kitaev chains and Ising models for $h = 0$. The Kitaev chain (left) can be gauged by following the procedure outlined in section 3.2. The resulting Hamiltonian is identified with a TFIM (center) through a *local* transformation. Yet a different TFIM (right) can be obtained either as the Jordan-Wigner dual of the original Kitaev chain, or as the Kramers-Wannier dual of the aforementioned Ising model (i.e., of the gauged Kitaev chain).

is the golden ratio[15], as made well-known by recent studies of the Rydberg chain [86–89]. As soon as we perturb $\mu > h$, the system naturally prefers a maximal number of spins to point down. Given that we have to satisfy the aforementioned constraint at low energies, the two possible ground states are the antiferromagnet ground states $|+-+-\rangle$ and $|-+-+\rangle$, giving us the deconfined phase. If instead $\mu < h$, the ground state is given by $|++++\rangle$, leading to confinement.

Starting from the degenerate point ($\mu = h$), it is also interesting to consider the effect of turning on $t \neq 0$. This term brings us out of the low-energy Hilbert space, but at leading order in $t$, we have the projected Hamiltonian

$$H_{\text{eff}} = 2\mu \sum_j \mathcal{P}_{j-1/2}\mathcal{P}_{j+1/2} + (\mu - h) \sum_j X_{j+1/2} - t \sum_j \mathcal{P}_{j-1/2}Z_{j+1/2}\mathcal{P}_{j+3/2}. \tag{34}$$

If we were to do a change of basis $X \leftrightarrow Z$, this can be recognized as the celebrated PXP model with its quantum scars [90, 91]. This is indeed known to have an Ising transition for $\mu - h \approx 0.7t$ [92, 93], or in other words, $h/\mu \approx 1 - 0.7\, t/\mu$, which sets the slope of the solid black line in Fig. 2(a) as it emerges from the vertical axis, agreeing with our numerical phase diagram.

We note that the effective constrained Hilbert space has a nice interpretation in terms of the original gauge theory (14): at the point $\mu = h$, $t = 0$, the energy cost of a pair of neighboring fermions (a dimer) is zero since the cost of flipping one electric string, as required by gauge-invariance, is exactly compensated by the gain due to the chemical potential $\mu$. In this language, the constrained Hilbert space is formed by all possible degenerate configurations of dimers of unit length. At small $t$, the physics of these dimers is governed by the PXP model.

## 3.8 The connection between gauging, Jordan-Wigner and Kramers-Wannier

Thus far, we have encountered the Ising chain in two different contexts. Firstly, it appeared in section 2.2 after the *nonlocal* Jordan-Wigner transformation of the Kitaev chain. Later, it appeared in section 3.4 as a *local* rewriting of the *gauged* Kitaev chain. However, clearly these two procedures are not the same, since the resulting Ising chains have their phases swapped: in Eq. 6 we saw that the symmetry-breaking phase occurs where $t$ is dominant, whereas in Eq. (19) it was where $\mu$ is dominant. Here we clarify these relationships.

Starting from the Kitaev chain (1) we saw that gauging fermion parity symmetry led to a symmetry-breaking deconfined phase, and a symmetry-protected topological Higgs phase.

---

[15]This follows from the observation that on a finite chain of length $L$ the number of states in the Hilbert space of the model is given by the Fibonacci number $F_L$.

Morever, the latter had fermionic edge modes and is thus a fermion SPT phase. This is summarized in the gray box in Fig. 3. Moreover, we saw that a local change of variables (18) mapped the latter SPT phase to a trivial product state, which is summarized by the second black arrow in Fig. 3. We discussed how this Ising chain has a magnetic $\mathbb{Z}_2$ symmetry for $h = 0$. In principle, this symmetry could also be gauged. We discuss this in detail in appendix B, where similar to before, we find that the trivial phase maps to a symmetry-breaking phase, and the other phase maps to a (now bosonic) SPT phase, as shown in Fig. 3. Again, a local change of variables can trivialize the latter. In effect, this ends up swapping the trivial and symmetry-breaking phases of the Ising chain, being equivalent to a Kramers-Wannier transformation. As summarized in Fig. 3, concatenating all these transformations is effectively equivalent to the Jordan-Wigner transformation encountered in section 2.2.

These relationships between gauging and the Jordan-Wigner and Kramers-Wannier transformations have been pointed out before in the continuum [43] and on the lattice [44]. However, in these cases, the subtlety of the local mappings was not addressed and the SPT phases were overlooked.

# 4 Gently gauging the Kitaev chain

As we demonstrated in section 3, gauging is a drastic operation that radically changes the physics of the Kitaev chain. It is natural to ask if the Kitaev model and its gauged counterpart can both emerge from a unified framework, where one can study the phase transition separating them. In addition, in section 3 we saw that the Higgs phase is topologically non-trivial with respect to the $\mathbb{Z}_2^f \times \mathbb{Z}_2$ symmetry. This might seem unusual, given that our fermion parity is gauged—the catch of course being that in the presence of a boundary this global symmetry actually survives gauging. To get a different perspective on this subtlety, it is valuable to see this SPT phase arise in an emergent gauge theory, where the Gauss law is not hardwired into the Hilbert space but is merely energetically implemented such that the fermion parity is truly a symmetry in the Hilbert space, even in the bulk.

For these reasons, in this section we will construct and analyze a theory which interpolates between the ordinary Kitaev and gauged Kitaev chains. To this purpose, consider the following Hamiltonian that acts in the unconstrained Hilbert space that includes link and site variables

$$H = \sum_j \left( it\tilde{\gamma}_j \sigma^z_{j+1/2} \gamma_{j+1} + i\frac{\mu}{2}\tilde{\gamma}_j \gamma_j - h\sigma^x_{j+1/2} - iK\sigma^x_{j-1/2}\tilde{\gamma}_j \gamma_j \sigma^x_{j+1/2} - \frac{t^2}{K}\sigma^z_{i+1/2} \right), \quad (35)$$

with a new parameter $K \geq 0$.

In the limit $K \to 0$, this model essentially reduces to the Kitaev chain (1): although the Hilbert space still contains degrees of freedom on the links, due to the last term in the Hamiltonian these are frozen to $\sigma^z = +1$ and thus completely decouple from the fermions. On the other hand, as $K \to \infty$ the next-to-last term in the Hamiltonian enforces a large energetic penalty to every state that does not satisfy the Gauss law. Hence in this limit, at energies much below the energy scale $K$, we recover the gauged model (14). For intermediate values of $K$, the Hamiltonian (35) interpolates between these two limiting regimes. In this paper we refer to this procedure as a gentle gauging of the Kitaev chain.

## 4.1 Quantum phase diagram and exact dualities in the absence of vortices

At $h = 0$ the model (35) enjoys a global $\mathbb{Z}_2^f \times \mathbb{Z}_2$ symmetry, generated by the fermionic parity $P = \prod_j \left( i\tilde{\gamma}_j \gamma_j \right)$ and the 'Wilson loop' $W = \prod_j \sigma^z_{j+1/2}$. We expect the phases of the gently

gauged model (35) to be classified in terms of these two symmetries.

We investigate the quantum phase diagram as a function of two dimensionless parameters $\mu/t$ and $K/t$. It is enough to consider only the interval $\mu, t \geq 0$, because at $h = 0$ the other regions are related by a unitary transformation. In addition to the two limits $K = 0$ and $K = \infty$ described above, the behavior of the model can also be understood exactly in the limit $\mu \to \infty$. In this case the fermionic bands are fully occupied: hopping and particle number fluctuations are therefore impossible, and the local fermion parity $i\tilde{\gamma}_i \gamma_i = -1$ everywhere. In that limit the Hamiltonian (35) at $h = 0$ reduces to

$$H = \sum_j K \sigma^x_{j-1/2} \sigma^x_{j+1/2} - \frac{t^2}{K} \sigma^z_{j+1/2}, \tag{36}$$

which is the TFIM (with the link variables being the degrees of freedom), exhibiting a phase transition from a disordered to the SSB phase at $K/t = 1$. For small $K$, this is the same trivial phase as we encountered in the Kitaev chain (1), whereas for large $K \to \infty$, the symmetry-breaking phase becomes the deconfined phase of the gauge theory discussed in section 3.5.

As a first step towards mapping out the quantum phase diagram of the gently gauged model, we apply a non-local transformation to the Hamiltonian (35). In particular, we introduce $\mathbb{Z}_2$ gauge-invariant Majorana operators on sites

$$\eta_{2i+1} = \left( \prod_{k<i} \sigma^z_{k+1/2} \right) \gamma_i, \qquad \tilde{\eta}_{2i+1} = \left( \prod_{k<i} \sigma^z_{k+1/2} \right) \tilde{\gamma}_i, \tag{37}$$

and also a new set of Majorana operators on links through the "hybrid" Jordan-Wigner transformation

$$\eta_{2i} = \left( \prod_{k<i} \sigma^z_{k+1/2} e^{i\pi n_{k+1}} \right) \sigma^x_{i+1/2}, \qquad \tilde{\eta}_{2i} = \left( \prod_{k<i} \sigma^z_{k+1/2} e^{i\pi n_{k+1}} \right) \sigma^y_{i+1/2}. \tag{38}$$

In terms of these, the model (35) takes the form

$$H = i \underbrace{\sum_j \left( t\, \tilde{\eta}_{2j-1} \eta_{2j+1} + \frac{\mu}{2} \tilde{\eta}_{2j+1} \eta_{2j+1} \right)}_{H_1} - i \underbrace{\sum_j \left( -K\, \tilde{\eta}_{2j} \eta_{2j+2} + \frac{t^2}{K} \tilde{\eta}_{2j} \eta_{2j} \right)}_{H_2}. \tag{39}$$

These are just two decoupled Kitaev chains governed by the Hamiltonians $H_1$ and $H_2$, whose phase diagram depends only on $\mu/t$ and $K/t$, respectively. For any value of $K$, the Hamiltonian $H_1$ is critical at $|\mu/t| = 2$. Conversely, the Hamiltonian $H_2$ is critical at $|K/t| = 1$ for any value of $\mu$. As a result, the phase diagram in the positive $\mu$-$K$ quadrant is divided into four rectangular regions by these two critical lines, as illustrated in Fig 4. Besides, as a consequence of the well known dualities of the Kitaev chains, each region can be exactly mapped onto one of the other three.

While this analysis allows us to correctly identify the phase boundaries, we need to refer to the original model (35) to understand the nature of the four discovered phases.[16] In phases I and II ($K < t$) the link spin fields form a trivial paramagnet, while the fermionic sector is smoothly connected to the pure Kitaev chain limit ($K = 0$) which undergoes a topological-to-trivial phase transition as $\mu/t$ is varied. Therefore, we label phase I as "Kitaev" and phase II as "Trivial". On the other hand, the nature of phase III can be inferred from the limit $\mu \to \infty$ governed by the Hamiltonian (36). For $K > t$ its ground state forms an Ising antiferromagnet,

---

[16]Under a non-local transformations, the physics of a quantum phase is generically modified. The paradigmatic example is the duality between the Kitaev chain and the TFIM, reviewed in section 2.2

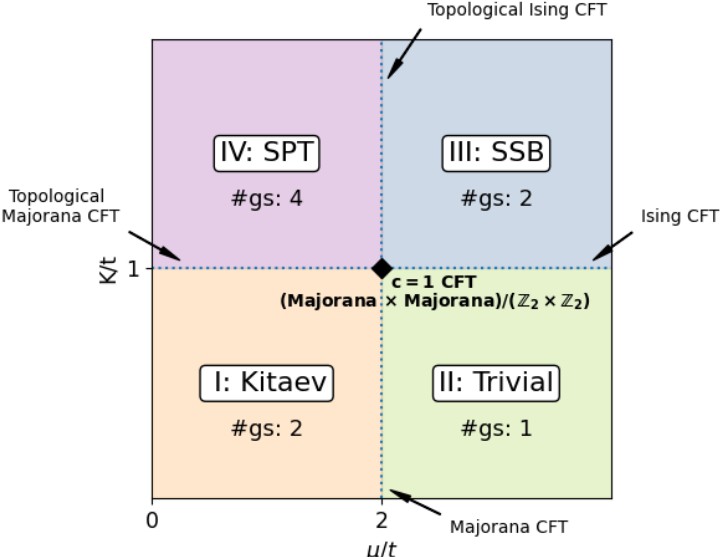

Figure 4: The phase diagram for the Hamiltonian (35) at $h = 0$. The transition lines are straight, as a consequence of two exact dualities explained in the main text. The intersection point (solid) corresponds to a conformal field theory with central charge $c = 1$. Ground state degeneracy in open chain geometry for each phase is also presented. The limit $K \to 0$ corresponds to the ordinary Kitaev chain (1) whereas $K \to +\infty$ is the gauged Kitaev chain (14) studied in section 3.

and therefore we refer to this phase as "Spontaneously Symmetry Broken (SSB)". As for phase IV, its SPT nature in the gauge limit $K \to \infty$ was proved in section 3.6. This fermionic SPT belongs to the same class as a stack of two Kitaev chains, which can be shown as follows.[17] Consider the limiting case $\mu = 0, K \gg t$, where we have the stabilizer code

$$H = \sum_j \left( it\tilde{\gamma}_j \sigma^z_{j+1/2} \gamma_{j+1} - iK \sigma^x_{j-1/2} \tilde{\gamma}_j \gamma_j \sigma^x_{j+1/2} \right). \tag{40}$$

Let us define the following Majorana modes, obtained from the original Majorana and link variables through a *local* transformation:

$$\begin{aligned}
\eta_{j,A} &= \gamma_j, \\
\tilde{\eta}_{j,A} &= \tilde{\gamma}_j \sigma^z_{j+1/2}, \\
\eta_{j,B} &= \tilde{\gamma}_j \sigma^x_{j+1/2}, \\
\tilde{\eta}_{j,B} &= \tilde{\gamma}_j \sigma^y_{j+1/2}.
\end{aligned} \tag{41}$$

Using these new variables, the Hamiltonian (40) reads

$$H = \sum_j \left( it\tilde{\eta}_{j,A}\eta_{j+1,A} - K(i\tilde{\eta}_{j,A}\eta_{j+1,A})(i\tilde{\eta}_{j,B}\eta_{j+1,B}) \right). \tag{42}$$

Despite being an interacting Hamiltonian, its ground state is a free-fermion state. Indeed, using the fact that $i\tilde{\eta}_{j,A}\eta_{j+1,A}$ is a local integral of motion, it is easy to see that for $K > 0$ the ground state does not change along the following path parametrized by $\lambda$:

$$H = \sum_j \left( it\tilde{\eta}_{j,A}\eta_{j+1,A} - (1-\lambda)K(i\tilde{\eta}_{j,A}\eta_{j+1,A})(i\tilde{\eta}_{j,B}\eta_{j+1,B}) + i\lambda\tilde{\eta}_{j,B}\eta_{j+1,B} \right). \tag{43}$$

---

[17]We emphasize that this is not guaranteed by Eq. (39), since the mapping (37)-(38) involves a non-local transformation.

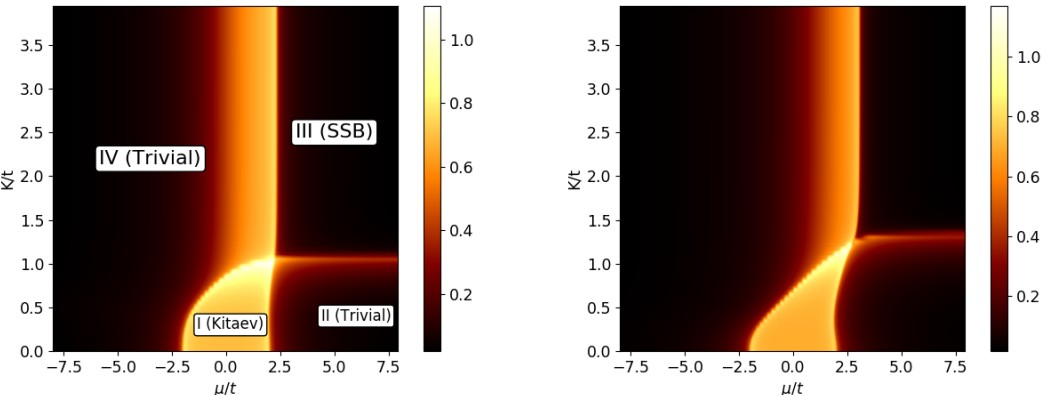

Figure 5: Half-chain entanglement entropy for a system described by the Hamiltonian (35) with $h/t = 1$ (left) and $h/t = 2$ (right). Results are obtained using DMRG for a chain of length $L = 100$. Phases are labeled as in section 4.1, except that the phase IV is not SPT anymore, but becomes trivial here. (For $h/t = 0$, see Fig. 4; for $h/t = 5$, see Fig. 6.)

While for $\lambda = 0$ this is the same as Eq. (42), for $\lambda = 1$ the Hamiltonian describes a stack of two Kitaev chains. Its ground state corresponds to an SPT phase protected either by complex conjugation, or by the $\mathbb{Z}_2^f \times \mathbb{Z}_2^f$ group of fermionic parities of each chain.

We have thus completely mapped the quantum phase diagram of the Hamiltonian (35) at $h = 0$, see Fig. 4. There are four distinct phases that are classified in terms of the global $\mathbb{Z}_2^f \times \mathbb{Z}_2$ symmetry of the model. The phases are separated by the two straight transition lines $K/t = 1$ and $\mu/t = 2$. The boundaries between different phases are critical. While all (except the multicritical point) are conformal field theories (CFTs) with central charge $c = 1/2$, they are all distinct. In particular, two are fermionic and two are bosonic: the two transitions out of the Kitaev phase are Majorana CFTs whereas the two transitions out of the Ising phase are Ising CFTs. Moreover, the two Majorana CFTs are topologically distinct: in the sense of Ref. [80] they are symmetry-enriched such that the transition between the Kitaev phase and the SPT phase is itself topologically non-trivial (with protected edge modes). Similarly, the Ising CFT between the SSB and SPT phases is also topologically non-trivial. These four critical lines meet at a multicritical point which is a CFT with central charge $c = 1$. It can be identified[18] with the field theory labeled $S_2$ in Fig. 2 of Ref. [46].[19]

## 4.2 Quantum phase diagram in the presence of vortices

In the presence of vortices ($h > 0$) the magnetic symmetry generated by $W = \prod_i \sigma_{i+1/2}^z$ is explicitly broken. Since our understanding of the quantum phases of the gently gauge model at $h = 0$ in Sec. 4.1 relied on such symmetry being present, we anticipate qualitative differences once a finite $h$ is turned on.

The physics of the Kitaev (I) and trivial (II) phases is essentially unchanged. This is clear from the fact that in the $K \to 0$ limit the $h$-term is negligible compared to the last term of the Hamiltonian (35), and moreover the fermion parity symmetry that characterizes the ordinary

---

[18]There are only three points in Fig. 2 of Ref. [46] that can be related to free fermions. One is the Dirac CFT, but this cannot be perturbed into an Ising CFT; another is the stack of a Majorana CFT and an Ising CFT, but in the phase diagram in Fig. 4 the Majorana CFTs make a 90° turn at the multicritical point, rather than being a straight line. By exclusion, we are dealing with the third option, the $S_2$ theory.

[19]Note that the transformation (39) maps the $c = 1$ multicritical point into a standard Dirac CFT, but this mapping is *non-local*.

Kitaev chain is still present.

The fate of the symmetry broken phase (III) depends on the sign of the chemical potential. This is clear from our discussion in Sec. 3, where we showed that at $K \to \infty$ the problem is governed by the asymptotic TLFIM Hamiltonian (19), whose Ising coupling is determined by the chemical potential. As explained in section 3.7, the Ising symmetry broken phase survives at finite longitudinal field only in the antiferromagnetic case. Therefore, we need to discuss the ferromagnetic ($\mu < 0$) and antiferromagnetic ($\mu > 0$) regimes separately. In the former case, the phase III becomes a trivial paramagnet with no ground state degeneracy as soon as a finite $h$ is introduced. In the latter case, SSB phase is still present at $h \neq 0$, but the nature of SSB is modified compared to the $h = 0$ problem. We note that one can the reach same conclusions about phase III at $h \neq 0$ by examining a different limit of the Hamiltonian (35), where $\mu \to \pm\infty$ but the coupling $K$ is finite. Here we get a different asymptotic TLFIM

$$H_{\mu=\pm\infty} = \sum_j \left( \pm K\, \sigma^x_{j-1/2}\sigma^x_{j+1/2} - \frac{t^2}{K}\sigma^z_{j+1/2} - h\sigma^x_{j+1/2} \right), \tag{44}$$

where the sign of the Ising term is still determined by the sign of the chemical potential $\mu$.

Finally, we consider phase IV: the absence of the $\mathbb{Z}_2$ magnetic symmetry destroys the SPT order and lifts the ground state degeneracy completely. We are left with another trivial phase.

The complete quantum phase diagram at $h \neq 0$ was mapped numerically and is shown in Fig. 5. For $\mu < 0$ there are only two regions: the Kitaev phase (I) is separated from a single trivial phase (IV) by the Majorana critical line. The case $\mu > 0$ is more interesting, as we still find four distinct quantum phases. Since phases II and IV are now both trivial, it is natural to ask why they are not connected. In other words, we want to understand why the special $c = 1$ critical point that separates them at $h = 0$ still survives at a finite $h$. In order to answer this question, we consider at first the large $h$ limit: $h \gg t, K, t^2/K, \mu$. In that regime we can replace $\sigma^x \to 1$, $\sigma^z \to 0$ and the Hamiltonian (35) takes the simple form

$$H_{h=\infty} = i \left( \frac{\mu}{2} - K \right) \sum_j \tilde{\gamma}_j \gamma_j = \left( \frac{\mu}{2} - K \right) \sum_j \left( 1 - 2n_j \right), \tag{45}$$

i.e. the fermionic sites are either completely occupied or completely empty depending on the sign of the prefactor. Remarkably, these are two distinct phases in the presence of translation symmetry. To see this, one can consider the string order parameter for fermion parity symmetry. Considering that it is an (unbreakable) symmetry, there will be long-range order of $\langle \mathcal{O}_i P_{i+1} \cdots P_{j-2}P_{j-1}\mathcal{O}_j \rangle$ for some appropriate choice of endpoint operator $\mathcal{O}_j$. Moreover, since parity is a $\mathbb{Z}_2$ symmetry, the momentum of this endpoint operator can only[20] be 0 or $\pi$. We thus have a discrete invariant. Moreover, the two fixed-point limits discussed above (where every site is empty or fully-occupied) realize both cases. They must thus be separated by a quantum critical point. We can think about these states as defining two distinct symmetry protected trivial (SPt) states [94] protected by the fermion parity $\mathbb{Z}_2^f$ symmetry and translation symmetry.

In the region of parameters specified above, one can use perturbation theory to find corrections to the simple Hamiltonian (45). We have to consider virtual processes induced by the full Hamiltonian (35), that move the states away and then back into the low-energy $h \to \infty$ Hilbert space, where $\sigma^x = 1$, $\sigma^z = 0$. At second order, we have one such process where one link is first flipped by the first term of the Hamiltonian (35) and then flipped back by the

---

[20]The endpoint operator of the square of the symmetry is the square of the endpoint operator. Since the endpoint operator of the trivial string has zero momentum, the momentum of the original endpoint operator has to satisfy $2k \equiv 0 \mod 2\pi$.

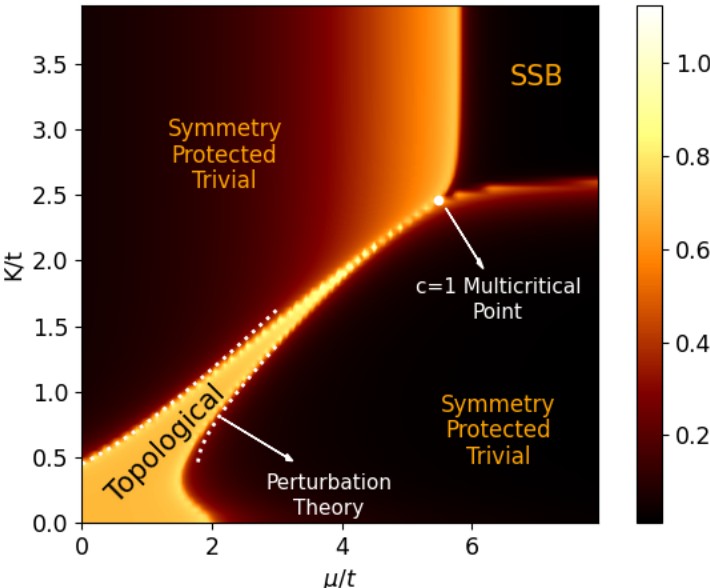

Figure 6: Numerical results for the half-chain entanglement entropy at $h/t = 5$. The white dotted lines are the analytical results from perturbation theory at large $h$, see Eq. (47). This approximation is valid for $t^2/h \ll K \ll h$ and $\mu \ll h$, where it reproduces correctly the phase boundaries. However, it cannot be used to infer that the critical lines converge into a $c = 1$ multicritical point.

last term, or vice versa. This gives a hopping contribution leading to the following effective fermionic Hamiltonian

$$H_{eff} = i\left(\frac{\mu}{2} - K\right)\sum_j \tilde{\gamma}_j \gamma_j - i\frac{t^3}{Kh}\sum_j \tilde{\gamma}_j \gamma_{j+1}\,. \tag{46}$$

This is a Kitaev chain, which is critical when

$$\frac{\mu}{2} - K = \pm\frac{t^3}{Kh}\,. \tag{47}$$

For a fixed $h$, the two positive solutions of this quadratic equation in the coupling $K$ give critical lines $K^+(\mu, t)$ and $K^-(\mu, t)$ which separate the topological Kitaev phase I from the trivial phases II and IV. As shown in Fig. 6, these lines agree well with our numerical results in the region of parameters described above, where perturbation theory is applicable. At $\mu = 0$, the transition happens for $K^* = (t^3/h)^{1/2}$. The two transition lines converge to each other without touching for large values of $\mu$. This provides additional evidence that the two trivial phases are indeed separated. Note however that the critical point with $c = 1$, where the topological region ends, lies outside the range $h \gg t, K, t^2/K, \mu$ for which Eq. (46) is a valid approximation. Therefore the critical point cannot be located with this method.

## 5 Gauging the fermion parity in a particle-conserving chain

Thus far, we have coupled the superconducting Kitaev chain to a $\mathbb{Z}_2$ gauge field (either exactly as in section 3 or gently as in section 4). Here we study what happens if we instead gauge a particle-conserving Hamiltonian, i.e., a one-dimensional Luttinger liquid. Despite the resulting $U(1)$-symmetric system being gapless, we will see that it still forms an SPT phase. More

generally, we will study the interpolation between this model and the gauged Kitaev chain; this leads to an interpretation of the gapless case as a topological phase transition between two distinct gapped SPT phases which are both non-trivial with respect to $\mathbb{Z}_2^f \times \mathbb{Z}_2^T$ symmetry.

There has already been a large interest in finding analogues of the topological Kitaev chain in the presence of particle number conservation. The corresponding models are gapless: sometimes these have algebraically-localized edge modes [95–100] whereas in other cases additional gapped degrees of freedom give rise to exponentially-localized edge modes [101–109]. The model we present in this work is of the latter type. In fact, our work shows that coupling to a gauge field is a new mechanism for creating such topological particle-conserving phases of matter.

## 5.1 The model: from Kitaev chain to particle conservation

Here we will study only the case with no vortices ($h = 0$). The model Hamiltonian is

$$H_\delta = i\, t(1-\delta) \sum_j \tilde{\gamma}_j \sigma^z_{j+1/2} \gamma_{j+1} - i\, t\delta \sum_j \gamma_j \sigma^z_{j+1/2} \tilde{\gamma}_{j+1} + \frac{i\mu}{2} \sum_j \tilde{\gamma}_j \gamma_j, \tag{48}$$

$$= -t \sum_j \left( c_j^\dagger \sigma^z_{j+1/2} c_{j+1} + h.c. \right) - (1-2\delta)t \sum_j \left( c_j^\dagger \sigma^z_{j+1/2} c_{j+1}^\dagger + h.c. \right) - \mu \sum_j \left( c_j^\dagger c_j - \tfrac{1}{2} \right), \tag{49}$$

where $0 \leq \delta \leq 1$. In addition, we impose the $\mathbb{Z}_2$ Gauss law

$$G_j = \sigma^x_{j-1/2} i\tilde{\gamma}_j \gamma_j \sigma^x_{j+1/2} = \sigma^x_{j-1/2} (-1)^{n_j} \sigma^x_{j+1/2} = 1. \tag{50}$$

One may choose to see this as a constraint hardwired into the Hilbert space (as in Section 3) or as being energetically imposed by an additional term in the Hamiltonian (as in Section 4). Note that for $\delta = 0$ we have the (gauged) Kitaev chain, whereas for $\delta = 1/2$ we have the (gauged) particle-conserving chain. In the latter case, matter has a full $U(1)$ symmetry, whereas we have gauged only its $\mathbb{Z}_2^f \subset U(1)$ subgroup; this model has been studied before in Ref. [40], although its topological properties and emergent anomalies—the focus of the present discussion—were not discussed.

For an open chain of $L$ sites that terminates with links, the Hamiltonian can be rewritten (up to a constant) using the mapping (18) as

$$H_\delta = -t \sum_{j=1}^{L-1} \left( (1-\delta) Z_{j+1/2} - \delta X_{j-1/2} Z_{j+1/2} X_{j+3/2} \right) + \frac{\mu}{2} \sum_{j=1}^{L} X_{j-1/2} X_{j+1/2}. \tag{51}$$

As before, the global fermion parity symmetry $P$ only acts non-trivially near the edge, $P = X_{1/2} X_{L+1/2}$, due to the Gauss law in the bulk. Moreover, the above spin model has the magnetic symmetry $W = \prod_j \sigma^z_{j+1/2} \propto P \prod_j Z_{j+1/2}$. The fact that the fractionalized symmetry $P$ (with $P_l = X_{1/2}$ and $P_r = X_{L+1/2}$) anticommutes with $W$ still ensures that all energy eigenstates are at least two-fold degenerate for any value of $\delta$. Let us also observe that the model is invariant under complex conjugation $T = K$, which acts as $(\sigma^x, \sigma^y, \sigma^z) \to (\sigma^x, -\sigma^y, \sigma^z)$ in the original variables and as $(X, Y, Z) \to (X, -Y, Z)$ in the new variables. This symmetry will play an important role in the following discussion.

## 5.2 Emergent anomaly and intrinsically gapless SPT order

In this section, we consider $\delta = 1/2$. As discussed in Ref. [40], the model is a gapless Luttinger liquid with central charge $c = 1$ if $|t| > |\mu|/2$. Here we show that, in addition, it is topologically non-trivial and has an emergent anomaly.

For this value of $\delta$, the model (48) has a $U(1)$ symmetry generated by $Q = \frac{1}{2}\sum_j n_j$. We normalized the operator such that a $2\pi$ rotation produces the fermion parity, i.e., we have $e^{2\pi i Q} = \prod_n P_n = P$. Since we are studying a gauge theory, this is a do-nothing transformation—at least in the bulk, as discussed before. Indeed, for any local bulk operator $\mathcal{O}_j$ in our theory, we have $e^{2\pi i Q}\mathcal{O}_j e^{-2\pi i Q} = \mathcal{O}_j$. More precisely, this is true if we consider the model (48) as an exact gauge theory; however, if the Gauss constraint is merely energetically enforced, then we say that $Q = \frac{1}{2}\sum_j n_j$ is properly normalized for the *low-energy* theory, where all local operators in the bulk are indeed parity-even.

It is instructive to write this same operator in the spin variables of the model (51) where we have removed all gauge redundancy, $Q = \frac{1}{4}\sum_j\left(1 - X_{j-1/2}X_{j+1/2}\right)$. We readily confirm that this has the correct normalization

$$e^{2\pi i Q} = e^{2\pi i \times \frac{1}{4}\sum_j\left(1 - X_{j-1/2}X_{j+1/2}\right)} = \prod_j\left(X_{j-1/2}X_{j+1/2}\right) = \prod_j X_{j+1/2}^2 = 1\,, \tag{52}$$

where we have presumed periodic boundary conditions. Remarkably, however, nonlocal operators can carry fractional charge. For example, consider a semi-infinite string of the magnetic symmetry

$$e^{2\pi i Q}\left(\cdots Z_{j-3/2}Z_{j-1/2}\right)e^{-2\pi i Q} \;=\; \cdots Z_{j-3/2}Z_{j-1/2} \times \underbrace{e^{4\pi i \times \frac{1}{4}X_{j-1/2}X_{j+1/2}}}_{=\cos(\pi)=-1}\,. \tag{53}$$

Hence, the semi-infinite string of $W$ has a non-trivial charge under a $2\pi$-rotation. This implies that there is a mutual 't Hooft anomaly between $U(1)$ and $W$. To see this, note that the definition of a 't Hooft anomaly is that the symmetry cannot be consistently gauged [110]. In the above case, if we would gauge the magnetic symmetry $W$, then the above semi-infinite string operator would become a local operator (since the symmetry string itself would become invisible). The resulting theory thus has local operators which carry fractional charge under $U(1)$. This prevents one from gauging the latter symmetry.[21] Thus, one cannot gauge both $U(1)$ and $W$ due to their mutual 't Hooft anomaly.

If the Gauss law is only energetically enforced, then the anomaly is *emergent* at low energies. In this case in the full Hilbert space, there are also local operators[22] that are charged under a $2\pi$-rotation and we see that the $Q$ operator is improperly normalized. Such emergent anomalies are very interesting since it was recently pointed out that they[23] imply that the ground state form an *intrinsically-gapless SPT phase* [62]. This is due to the emergent anomaly being intimately related to long-range order in a so-called *impossible* string order parameter, i.e., one that is not allowed in a gapped symmetric phase of matter. In this case, this is the string order of fermion parity symmetry: $\cdots P_{j-1}P_j\sigma_{j+1/2}^x$. This is charged under $W$ (which, as before, protects edge modes). In and of itself, this does not seem 'impossible'—indeed, we encountered it in the gapped symmetric gauged Kitaev chain studied in Sections 3 and 4. However, it is in fact impossible if fermion parity symmetry is enhanced to a full $U(1)$ symmetry. To see this, suppose one has a gapped symmetric phase with $U(1)$ and $W$ symmetry. Then for every choice of $\alpha$, there exists a local operator $\mathcal{O}_j^{(\alpha)}$ such that $e^{i\alpha\sum_{k<j}n_j}\mathcal{O}_j^{(\alpha)}$ has long-range order. Since $W$ is a $\mathbb{Z}_2$ symmetry, this endpoint operator is either odd or even under $W$. At $\alpha = 0$, we can clearly choose $O_j^{(0)} = 1$, which is even under $W$. Since its *discrete* charge under $W$ cannot change as we smoothly vary $\alpha$, we conclude that for $\alpha = 2\pi$, the fermion parity string must

---

[21]Mathematically, one is effectively trying to gauge the quotient group of a bigger group, which one cannot consistently do. Of course, one way around it is to instead redefine $U(1)$ to make the fractional charge the unit charge. Indeed, extending symmetries allows to lift anomalies in general [111].

[22]This is also true in the case with a strict Gauss law constraint if we include boundary operators; see Section 3.4.

[23]This only applies for emergent anomalies of on-site symmetries; note that $Q$ is indeed on-site in the Hilbert space of the gently gauged model.

have a trivial endpoint operator, making long-range order in $\cdots P_{j-1}P_j\sigma^x_{j+1/2}$ impossible. It can only have long-range order in a gapless system (or in a phase that spontaneously breaks $W$, as happens for $|\mu|/2 > |t|$).

We thus conclude that for $\delta = 1/2$, the model (48) has an (emergent) anomaly and forms an intrinsically gapless SPT phase. In fact, the effective spin chain (51) is a well-known example of system with an anomalous $\mathbb{Z}_2$ symmetry [112, 113]. To make this connection, define $U = We^{i\pi Q}$. This indeed squares to the identity operator, at least in the sector satisfying the Gauss law. However, similar to Eq. (53), one can show[24] that the semi-infinite string of this symmetry is charged under $U^2$, implying that this $\mathbb{Z}_2$ symmetry is anomalous. When the Gauss law is enforced energetically, we indeed see that this symmetry actually defines a non-anomalous $\mathbb{Z}_4$ symmetry which becomes an effective anomalous $\mathbb{Z}_2$ symmetry at low energies, which is similar to the Ising-Hubbard chain discussed in Ref. [62].

Thus far, we have discussed how this gapless system is anomalous for $U(1) \times \mathbb{Z}_2$ (generated by $Q$ and $W$) and $\mathbb{Z}_2$ (generated by $We^{i\pi Q}$). It is worth noting that it is also anomalous for $U(1) \rtimes Z_2^T$, generated by $Q$ and $WT$. Indeed, note that the latter flips the sign of $\sigma^x_{j+1/2}$, such that the string order parameter for fermion parity symmetry is charged under it. As discussed, this is an impossible string order parameter if fermion parity is enhanced to $U(1)$; as explained in Ref. [62], this in turn implies an emergent anomaly at low energies.

### 5.3 Topological phase transition between distinct fermionic SPT phases

We now consider $\delta \neq 1/2$. Starting from the gapless case (i.e., $\delta = 1/2$ with $|t| > |\mu|/2$), the superconducting term immediately opens a gap. The regime $\delta < 1/2$ is adiabatically connected to $\delta = 0$, i.e., the gauged Kitaev chain that we have studied in the previous sections. In particular, this is a non-trivial SPT phase protected by $\mathbb{Z}_2^f \times \mathbb{Z}_2$ generated by $P$ and $W$. In fact, the same statement holds for $\delta > 1/2$. To see this, note that if we define $H_\delta(\alpha) = e^{i\alpha Q}H_\delta e^{-i\alpha Q}$, then $H_\delta(\pi) = H_{1-\delta}$. Since, $e^{i\alpha Q}$ commutes with $P$ and $W$, we have an adiabatic path of symmetric Hamiltonians connecting the region $\delta < 1/2$ with $\delta > 1/2$. Hence, from the perspective of this symmetry group, they form the same SPT phase.

However, this path does not preserve complex conjugation symmetry. Indeed, we now show that the two gapped regions are in fact distinct SPTs phases if we also preserve $T$. To see that, we return to fractionalization of the magnetic symmetry $W$ on a finite chain. We remind the reader that in the spin language the magnetic symmetry is given by $W \propto Y_{1/2}Z_{1+1/2}Z_{2+1/2}$ $\cdots Z_{L-1/2}Y_{L+1/2}$. As we have already argued in Sec. 3.6, in the limit $\delta \to 0$ and $\mu \to 0$ we can replace all $Z$ operators inside the string by unity and get the fractionalized form $W \propto Y_{1/2}Y_{L+1/2}$. On the other hand, as we take $\delta \to 1$ and $\mu \to 0$, we must instead replace $Z_{j+1/2} \to -X_{j-1/2}X_{j+3/2}$ and we thus end up with $W \propto Z_{1/2}Z_{L+1/2}$. In both cases the parity and magnetic symmetries are realized projectively at the edges, so we indeed deal with SPT phases. Importantly, however, the edge magnetic symmetry operators transform differently under time-reversal symmetry $T = K$. While in the former case at the left edge $TW_lT^{-1} = -W_l$, in the latter $TW_lT^{-1} = +W_l$. Since the transformation property cannot change gradually, we conclude that the two SPT phases are different, meaning that they cannot be connected without a phase transition along a trajectory that preserves relevant symmetries $P$, $W$ and $T$.

In fact, the two gapped phases already form distinct SPT phases for the smaller symmetry group $\mathbb{Z}_2^f \times \mathbb{Z}_2^T$ generated by $P$ and $WT$. To distinguish them with just this symmetry group, we study the edge mode operators in a fine-tuned limit. For $\delta = \mu = 0$, we know that the system has (gauge-invariant) edge mode operators $\sigma^z_{1/2}\gamma_1$ and $\sigma^x_{1/2}$. Equivalently, $\alpha_L = \sigma^z_{1/2}\gamma_1$ and $\beta_L = \sigma^y_{1/2}\gamma_1$. Note that both are hermitian Majorana operators and both commute with $WT$.

---

[24]Note that the semi-infinite string $\prod_{k \leq j}\sigma^z_{k-1/2}e^{i\pi n_k/2}$ is does not commute with the Gauss operator, hence one has to consider,, e.g., $\left(\prod_{k \leq j}\sigma^z_{k-1/2}e^{i\pi n_k/2}\right)\gamma_j$, which is indeed odd under $U^2 = P$.

In particular, this means that the edge perturbation gapping out the edge mode, $i\alpha_L\beta_L$, is forbidden by $WT$; hence, $WT$ indeed protects the SPT phase. But we can be more precise: let us define the complex edge mode operator $c_L = \alpha_L + i\beta_L$. We see that our anti-unitary symmetry $WT$ maps this to $(WT)c_L(WT) = c_L^\dagger$. This implies that $WT$ squares to $+1$ on this left edge. To derive that, let $|0\rangle_L$ be the vacuum of $c_L$ (i.e., $c_L|0\rangle_L = 0$). Moreover, define $|1\rangle_L = c_L^\dagger|0\rangle_L$. It is not hard to see that $WT|0\rangle_L = \rho|1\rangle$ (for some complex phase $\rho \in S^1 \subset \mathbb{C}$). Now,

$$(WT)^2|0\rangle_L = WT\rho|1\rangle = WT\rho c_L^\dagger|0\rangle_L = \bar{\rho}c_L WT|0\rangle_L = \bar{\rho}c_L\rho|1\rangle_L = c_L c_L^\dagger|0\rangle_L = |0\rangle_L, \quad (54)$$

as claimed. Similarly, since $(PWT)c_L(PWT) = -c_L^\dagger$, we see that $PWT$ squares to $-1$ on the left edge. In other words, this symmetry protects a zero-energy Kramers pair.

We can repeat this on the right edge, where we have the edge mode operators $\alpha_R = \tilde{\gamma}_N \times \sigma_{N+1/2}^z$ and $\beta_R = \tilde{\gamma}_N\sigma_{N+1/2}^y$. Now $WT$ negates both, so if we define $c_R = \alpha_R + i\beta_R$, then $(WT)c_R(WT) = -c_R^\dagger$. Repeating the above, we now derive that $(WT)^2|0\rangle_R = -|0\rangle_R$; the right edge mode has a Kramers pair for $WT$. Carrying out the same analysis for the $\delta = 1$ case, one finds the inverted case: now the left edge mode has a Kramers pair for $WT$ and the right edge mode has a Kramers pair for $PWT$.

The two distinct SPTs can be identified with the rows $\alpha = 2$ and $\alpha = -2$ in Table I of Ref. [82]. In other words, we can identify $\delta < 1/2$ as being the phase created by a stack of two Kitaev chains, whereas $\delta > 1/2$ is a stack of two spatially-inverted Kitaev chains. This is consistent with our explicit mapping of (48) with $\delta = 0$ to two decoupled Kitaev chains using the local change of variables (41).

In summary, the model (48) realizes two distinct non-trivial gapped SPT phases protected by the $\mathbb{Z}_2^f \times \mathbb{Z}_2^T$ symmetry. They are separated by a quantum critical point where $\mathbb{Z}_2^f$ is enhanced to $U(1)$. This critical point is itself topologically non-trivial, which is in turn intimately related to its anomaly for $U(1) \rtimes \mathbb{Z}_2^T$.

# 6 Numerical methods

We study the gently gauged model (35) numerically using both finite and infinite DMRG with the help of the tensor network Python library TeNPy [114]. Matrix Product States (MPS)-based DMRG operates within a bosonic/spin Hilbert space. Therefore, we apply a Jordan-Wigner transformation to the fermionic operators in the Hamiltonian (35) and work with the gently gauged version of the TFIM. Since the mapping is exact but non-local the two models exhibit similar quantum phase diagrams with identical phase boundaries, but with different interpretations of quantum phases and critical lines. In appendix B we investigate the gently gauged TFIM and discuss the main differences compared to the gently gauged Kitaev model.

The DMRG algorithm returns the ground state of a given Hamiltonian as a matrix product state (MPS). This gives direct access to the reduced density matrix of any subsystem and consequently to the mutual entanglement entropy of two subblocks of the chain under a bipartition. This quantity is particularly handy to locate the boundaries between distinct gapped phases, since it is known to diverge at the critical lines, where the system becomes gapless in the thermodynamic limit. For a finite system the entanglement entropy is also finite, but it is possible to extract information about the thermodynamic limit by increasing the system size and performing an extrapolation. In practice, we investigated chains of length $L \approx 100$ and detected the phase boundaries from peaks in the entanglement entropy. In particular, this was done to explore the quantum phase diagram at finite $h$, see Figs. 5, 6. In all cases the boundaries are clearly visible and agree with our analytical arguments. Moreover, the ground state degen-

eraries presented in Figs. 4, 8 were confirmed numerically by doing exact diagonalization on open chains of length $L \approx 15$.

# 7 Conclusion and outlook

There are at least two main take-away messages in this work.

Firstly, although gauging a global symmetry completely eliminates it in the bulk, near the edges the symmetry can meaningfully survive and can enrich the quantum phase diagram of the system. While in this paper we concentrated solely on gauging of the $\mathbb{Z}_2^f$ fermion parity in the one-dimensional Kitaev chain, these ideas naturally generalize to more complicated gauge groups and higher dimensions. This will be explored further in an upcoming paper [83].

Secondly, it can be instructive to interpolate between the gauged and ungauged model, which we refer to as gentle gauging. For instance, it makes the SPT phase of the Higgs condensate completely unambiguous, since in this emergent gauge theory we still have the 'gauge symmetry' as a true symmetry of the full microscopic Hilbert space. Moreover, we saw that the quantum criticality separating these distinct phases can have a rich phenomenology, with topologically distinct versions of the same underlying universality class meeting at a multicritical point which itself can be unusual—such as the $S_2$ criticality at the center of Fig. 4 where a bosonic Ising transition meets a fermionic Majorana transition.

In conclusion, gauging one of the simplest of symmetries—the fermion parity symmetry—in one of the most elementary of models—the Kitaev chain or the fermion hopping chain—can still have surprises in store (including a new mechanism to construct intrinsically gapless SPT phases). It would be interesting to extend this approach to fermionic systems in higher dimensions. For example, one can can investigate gauging of $\mathbb{Z}_2^f$ fermion parity of the $p + ip$ lattice superconductor and study the interplay between the edge global fermion parity symmetry and the $\mathbb{Z}_2$ magnetic one-form symmetry. More generally, we are hopeful that the concepts of SPT phases in Higgs condensates and of gentle gauging will prove to be useful for future works. *Note added:* After our preprint appeared, we became aware of an investigation of the quantum phase diagram of a gently-gauged one-dimensional abelian $U(1)$ lattice gauge theory [115].

# Acknowledgements

We acknowledge fruitful discussions with Abhinav Prem, Ryan Thorngren and Carl Turner. Our work is funded by the Deutsche Forschungsgemeinschaft (DFG, German Research Foundation) under Emmy Noether Programme grant no. MO 3013/1-1 and under Germany's Excellence Strategy - EXC-2111 - 390814868, by the Harvard Quantum Initiative Postdoctoral Fellowship in Science and Engineering (RV) and a grant from the Simons Foundation (#376207, Ashvin Vishwanath) (RV).

# A Alternative derivation of the spin Hamiltonian (19)

We show here that the mapping that transforms the Hamiltonian (14) into a local spin model (19) can be seen as a combination of a Jordan-Wigner (JW) and a Kramers-Wannier (KW) transformations. While the local mapping (18) is more elegant and completely avoids subtleties related to the non-locality of the JW and KW transformations, the alternative approach is instructive and worth presenting here.

First, we express the electric operator $\sigma^x$ by iteratively resolving the gauge constraint

$$\sigma^x_{j-1/2} = (-1)^{n_j}\sigma^x_{j+1/2} = \prod_{i \geq j}(-1)^{n_i}. \tag{55}$$

This removes $\sigma^x$ from the Hamiltonian (14), at the price of introducing a non-local interaction between fermions. The Hamiltonian (14) still has a dependence on $\sigma^z$ in the hopping term. However, this can be eliminated by defining new non-local, $\mathbb{Z}_2$ gauge-invariant fermionic operators

$$f_i^\dagger = \prod_{j \leq i}\sigma^z_{j-1/2}c_i^\dagger, \tag{56}$$

in terms of which we have

$$c_i^\dagger \sigma^z_{i+1/2}c_{i+1} = f_i^\dagger f_{i+1}, \qquad c_i^\dagger \sigma^z_{i+1/2}c_{i+1}^\dagger = f_i^\dagger f_{i+1}^\dagger. \tag{57}$$

After the JW transformation applied to the $f$ fermions, we can express the Hamiltonian in terms of spin variables residing on sites only:

$$H = -t\sum_i \tilde{X}_i\tilde{X}_{i+1} - h\sum_i\prod_{j \leq i}\tilde{Z}_j - \mu\sum_i\frac{1-\tilde{Z}_i}{2}. \tag{58}$$

After the further KW transformation

$$X_{i+1/2} = \prod_{j \leq i}\tilde{Z}_j, \qquad Z_{i+1/2} = \tilde{X}_i\tilde{X}_{i+1}, \tag{59}$$

is done, one can see explicitly that the Hamiltonian becomes local. Up to a constant, one finds

$$H = -t\sum_i Z_{i+1/2} - h\sum_i X_{i+1/2} + \frac{\mu}{2}\sum_i X_{i-1/2}X_{i+1/2}, \tag{60}$$

which agrees with Eq. (19) in the main text.

# B Gauging the transverse-field Ising chain

In this Appendix we summarize how ideas developed in this paper can be applied to gauging of the $\mathbb{Z}_2$ Ising symmetry of the transverse-field Ising model (TFIM) in one spatial dimension. Salient features of the quantum phase diagram of the gauged ferromagnetic TFIM have been already discussed in Appendix B of Ref. [116]. Our analysis here will uncover new properties of the model which have not been fully appreciated before. We will also emphasize main differences between the gauged TFIM and the gauged Kitaev chain that was investigated in the main part of the paper.

The TFIM Hamiltonian is given by

$$H = -J\sum_j \tau_j^x\tau_{j+1}^x - f\sum_j \tau_j^z, \tag{61}$$

where $\tau_j^x$ and $\tau_j^z$ are Pauli matrices acting on sites of the chain. This Hamiltonian commutes with the spin flip operator $Q = \prod_j \tau_j^z$ which generates a global $\mathbb{Z}_2$ symmetry. In the spirit of section 3 we will now gauge this symmetry: first we enlarge the Hilbert space by Ising variables defined on links of the chain and denote Pauli operators acting on the links by $\sigma^i_{j+1/2}$ with $i = x, y, z$. Next, we impose the Gauss law constraint $G_j = \sigma^x_{j-1/2}\tau_j^z\sigma^x_{j+1/2} = 1$ which



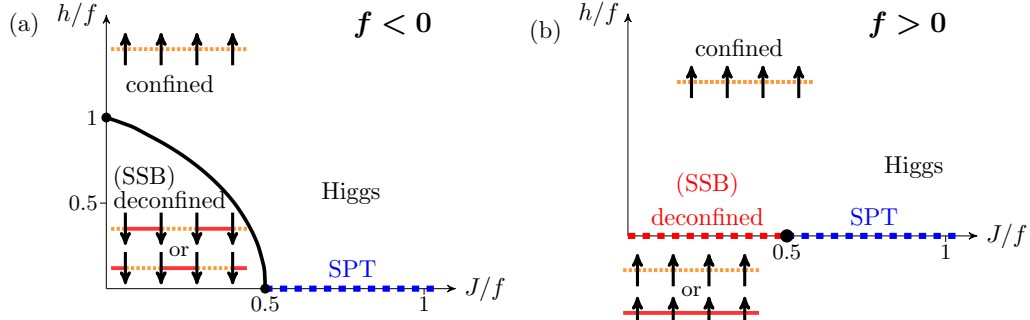

Figure 7: Phase diagram of the gauged TFIM (62) for $f < 0$ and $f > 0$, respectively. In absence of vortices ($h = 0$), the system enjoys the magnetic symmetry $W = \prod_j \sigma^z_{j+1/2}$, protecting the SPT order in the Higgs phase (highlighted by dashed blue line). For $h \neq 0$, the Higgs and confined regimes are adiabatically connected. For $f < 0$ the solid black line denotes Ising criticality, which is stabilized due the translation-breaking nature of the antiferromagnetic phase, similar to the discussion in section 3.7.

shrinks the physical Hilbert space and ties the Ising matter to the $\mathbb{Z}_2$ gauge fields. After using the minimal coupling prescription and introducing the kinetic electric term for the gauge fields, we end up with the Hamiltonian of the gauged TFIM

$$H = -J \sum_j \tau^x_j \sigma^z_{j+1/2} \tau^x_{j+1} - f \sum_j \tau^z_j - h \sum_j \sigma^x_{j+1/2}. \tag{62}$$

On a closed chain, the Gauss law implies that the Ising charge $Q$ must evaluate to unity in the Hilbert space of physical states. As a result, the global Ising $\mathbb{Z}_2$ symmetry is completely eliminated by gauging. On the other hand, on an open chain which terminates with links, the product of Gauss operators $G_j$ at all sites $j = 1, \ldots, L$ implies

$$Q = \sigma^x_{1/2} \sigma^x_{L+1/2}. \tag{63}$$

Hence similar to the fermion parity of the gauge Kitaev chain, here the Ising symmetry survives only at edges, where it fractionalizes.

We will now introduce the *gauge-invariant* spin variables

$$X_{j+1/2} = \sigma^x_{j+1/2}, \qquad Y_{j+1/2} = \tau^x_j \sigma^y_{j+1/2} \tau^x_{j+1}, \qquad Z_{j+1/2} = \tau^x_i \sigma^z_{j+1/2} \tau^x_{j+1}, \tag{64}$$

in terms of which the gauged Hamiltonian (62) on a closed chain can be written as

$$H = -f \sum_j X_{j-1/2} X_{j+1/2} - J \sum_j Z_{j+1/2} - h \sum_j X_{j+1/2}, \tag{65}$$

where in the first term we used the Gauss law and replaced $\tau^z_j \to X_{j-1/2} X_{j+1/2}$. The resulting Ising model in transverse and longitudinal fields is identical in the bulk to the gauge-invariant formulation of the gauged Kitaev chain (19) after the redefinitions of the coupling constants $f \to -\mu/2$ and $J \to t$. As a result, the quantum phase diagram of the gauged TFIM presented in Fig. 7 closely resembles Fig. 2. Note however that the properties of some quantum phases of the two models differ substantially, as we are going to highlight below.

At $h = 0$, the gauged TFIM exhibits symmetry-protected *bosonic* edge modes in the Higgs phase. One can construct these modes in the following way: On an open chain of length $L$

which starts and ends with links the definition of the gauge-invariant spins (64) cannot be applied to the outer left and right links, but instead we define

$$X_{1/2} = \sigma_{1/2}^x, \qquad Y_{1/2} = \sigma_{1/2}^y \tau_1^x, \qquad Z_{1/2} = \sigma_{1/2}^z \tau_1^x;$$
$$X_{L+1/2} = \sigma_{L+1/2}^x, \qquad Y_{L+1/2} = \tau_L^x \sigma_{L+1/2}^y, \qquad Z_{L+1/2} = \tau_L^x \sigma_{L+1/2}^z. \qquad (66)$$

Notice that in contrast to the gauged Kitaev chain, all edge gauge-invariant operators are bosonic and thus a priori can appear as individual terms in the Hamiltonian of an open chain. In the absence of vortices at $h = 0$, however, the model, in addition to the Ising symmetry (63), enjoys the magnetic symmetry $W = \prod_j \sigma_{j+1/2}^z = Q \prod_j Z_{j+1/2}$. As a result, all edge terms (66) are ruled out by symmetries. In particular, the Ising symmetry prohibits the edge $Y$ and $Z$ operators to appear in the Hamiltonian, while the magnetic symmetry does not allow $X$ and $Y$. As a result, at $h = 0$ the open chain Hamiltonian is

$$H = -f \sum_{j=1}^{L} X_{j-1/2} X_{j+1/2} - J \sum_{j=1}^{L-1} Z_{j+1/2}. \qquad (67)$$

We will now identify two edge operators localized near the left boundary that commute with this Hamiltonian. First, we have $X_l = X_{1/2}$. In addition, the operator

$$Y_l = Y_{1/2} + \frac{f}{J} Z_{1/2} Y_{3/2} + \frac{f^2}{J^2} Z_{1/2} Z_{3/2} Y_{5/2} + \cdots, \qquad (68)$$

also commutes with the Hamiltonian (67) and is exponentially localized near the left boundary in the Higgs phase, where $|f| < J$. The presence of two anti-commuting localized edge operators $X_l$ and $Y_l$ ensures two-fold ground state degeneracy associated with the left boundary. Since similar arguments apply also to the right edge, the total degeneracy of the ground state manifold on an open chain is four-fold with exponentially small corrections in system size $L$. The existence and stability of this degeneracy originates from fractionalization of the Ising and magnetic symmetries. In particular, the two $\mathbb{Z}_2$ symmetries anti-commute with each other at each edge and thus are realized projectively at the boundary.

To gain addition insight into the nature of the SPT phase, in the rest of this Appendix we will investigate the gently gauged TFIM at $h = 0$. Its Hamiltonian is given by

$$H = \sum_j \left( -J \, \tau_j^x \sigma_{j+1/2}^z \tau_{j+1}^x - \frac{J^2}{K} \sigma_{j+1/2}^z - K \sigma_{j-1/2}^x \tau_j^z \sigma_{j+1/2}^x - f \, \tau_j^z \right), \qquad (69)$$

with $K > 0$. By construction the model interpolates between the ordinary TFIM (61) as $K \to 0$ and the gauged TFIM (62) (with $h = 0$) in the limit $K \to \infty$. The model (69) enjoys a global $\mathbb{Z}_2 \times \mathbb{Z}_2$ symmetry, generated by $Q = \prod_j \tau_j^z$ and $W = \prod_j \sigma_{j+1/2}^z$. Since the site and link variables appear in such a symmetric way in Eq. (69), we can define a new lattice with new sites placed at sites and links of the original chain and rewrite the Hamiltonian as

$$H = \sum_a \left( \lambda^a \, \tilde{\tau}_{a-1}^x \tilde{\tau}_a^z \tilde{\tau}_{a+1}^x - g^a \, \tilde{\tau}_a^z \right), \qquad (70)$$

where $\tilde{\tau}_a^x$ and $\tilde{\tau}_a^z$ denote Pauli matrices acting on sites $a$ of the new lattice. In addition, $\lambda^a$ and $g^a$ are space-dependent couplings that take different values on odd and even sites of the new lattice. The Hamiltonian (70) defines a cluster model in an external field with couplings that alternate in space.

The quantum phase diagram of the gently gauged model (69) contains four distinct phases, as illustrated in Fig. 8. Since this model can be mapped by a Jordan-Wigner transformation to

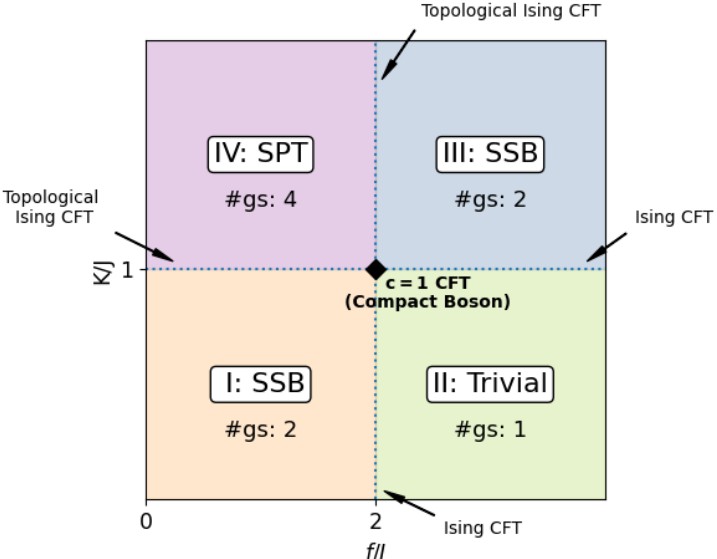

Figure 8: Quantum phase diagram of the gently gauged TFIM (69) at $h = 0$. The Ising transition lines are straight, as a consequence of the exact dualities. The intersection point (diamond) corresponds to a CFT with central charge $c = 1$.

the gently gauged Kitaev model of section 4, the critical lines are the same in the two cases. In the following we will emphasize how the phase diagram here differs from the phase diagram of the gently gauged Kitaev model presented in Fig. 4.

Consider first the limit $f \to 0$, $K \to \infty$, belonging to the SPT region IV in Fig. 8. In this limit both external field terms drop out, and we end up with the cluster model [117–121]. The space dependence of the cluster couplings does not matter, since each term in the cluster Hamiltonian commutes with the others. The cluster model realizes an SPT phase with four-fold ground state degeneracy, protected by the global $\mathbb{Z}_2 \times \mathbb{Z}_2$ symmetry identified above. The cluster state can be written in the matrix-product form with the bond dimension $D = 2$ [122]. The Schmidt values of a bipartition $\lambda_1 = \lambda_2 = 1/\sqrt{2}$ result in the entanglement entropy $S = \log 2$, which is in agreement with our numerical findings deep in the SPT region IV. It is natural that in contrast to the fermionic SPT order that we established for the phase IV of the gently gauged Kitaev model at $h = 0$, here the SPT phase has bosonic nature.

Another important difference compared to our analysis of the gently gauged Kitaev chain is that phase I displayed in Fig. 8 is not topological here, but exhibits spontaneous symmetry breaking of the Ising $\mathbb{Z}_2$ symmetry generated by $Q$. This of course is consistent with the well-known statement that the Kitaev chain and the Ising chain are related by a Jordan-Wigner transformation.

The critical point at $J = K = f$ has a simple interpretation within the spin model (70). Here all couplings are the same and one gets

$$H = -J \sum_a \tilde{\tau}_a^z \left( \tilde{\tau}_{a-1}^x \tilde{\tau}_{a+1}^x + 1 \right). \tag{71}$$

After the unitary transformation $U = \dots \tilde{\tau}^z \tilde{\tau}^z \tilde{\tau}^0 \tilde{\tau}^0 \tilde{\tau}^z \tilde{\tau}^z \tilde{\tau}^0 \tilde{\tau}^0 \dots$, the Hamiltonian transforms to

$$H = J \sum_a \tilde{\tau}_a^z \left( \tilde{\tau}_{a-1}^x \tilde{\tau}_{a+1}^x - 1 \right). \tag{72}$$

This spin model has a global $U(1)$ symmetry corresponding to the conservation of the total number of domain walls. It is dual to the particle-number conserving model of free hopping

fermions at half filling [40, 123], which explains why at this point the system is critical with the central charge $c = 1$. The critical point has two relevant deformations [124]: one gives rise to the Landau-forbidden quantum phase transition (a $1 + 1d$ deconfined quantum critical point [125]) between the two symmetry-broken phases I and III, while another leads to the topological phase transition between the trivial phase II and the bosonic SPT phase IV.

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
