# Peer review of "Gauging the Kitaev chain"

_SciPost Physics, doi:SciPost Phys. 10, 148 (2021)_

## Round 2 · Referee Report · Paul Fendley · 2021-4-11

Report

The authors carefully work out the consequences of gauging the fermion-parity symmetry of the Kitaev chain, including the effect of including a "magnetic"-symmetry breaking (i.e. vortex creating term) term. They show how it maps onto an effective Ising chain, with the vortex term becoming the longitudinal magnetic field. They extend their results to a model with an enhanced U(1)-fermion number symmetry, explaining the role of an anomaly.

I recommend publication in SciPost. While this is basic stuff, that is a virtue: the field will benefit from such a clear presentation and thorough treatment. Moreover, the authors provide an engaging discussion of how to physically interpret the different phases arising, including the appearance of a fermionic SPT phase. I am happy that the authors took the time.

I don't have any major suggestions. Perhaps the authors should mention that their models are all free-fermionic except when $h\ne 0$. Perhaps also the authors should stress a little more why the deconfined phase is stable only for $\mu<0$ (antiferromagnetic coupling in the effective Ising picture). The simplest reason I know is that translation symmetry plays the role of the Ising $\mathbb{Z}_2$ symmetry, which makes the longitudinal magnetic field irrelevant.

  • validity: -
  • significance: -
  • originality: -
  • clarity: -
  • formatting: -
  • grammar: -

Author:  Umberto Borla  on 2021-05-21  [id 1450]

(in reply to Report 1 by Paul Fendley on 2021-04-11)

We thank Prof. Fendley for the positive report. We have addressed the following comments:

  • Perhaps the authors should mention that their models are all free-fermionic except when $h \neq 0$.

We have added a sentence at the end of section 3.2, explaining that the case $h=0$ can be non-locally mapped to a free-fermion problem.

Change in manuscript: Added the sentence “Moreover, let us mention that although the case $h = 0$ can mapped to a free-fermion problem, this requires a nonlocal mapping which can obscure some of the relevant physics, so we do not follow this route.”

  • Perhaps also the authors should stress a little more why the deconfined phase is stable only for $μ<0$ (antiferromagnetic coupling in the effective Ising picture).

This is explained in some detail in section 3.7. We have added a sentence in the introduction that summarizes the argument.

Change in manuscript: Modified a sentence in section 1 to “In particular, after gauging, the trivial phase of the Kitaev chain becomes a deconfined phase which for a positive chemical potential is stable to vortices, despite matter being massive, due to spontaneous translation symmetry-breaking giving rise to deconfined domain walls.”

---

## Round 2 · Referee Report · Anonymous · 2021-4-19

Strengths

1- Gauging the Kitaev chain in the presence of boundary.

2- Determining the gauge structure of the fermion number in the bulk and at the boundary.

3- Determining the phase structure of the Kitaev chain as a function of the coupling strengths.

Weaknesses

1) Leaving a (possibly relevant) paper out.

2) Leaving certain quantities vague.

3) Gentle gauging of the Kitaev chain.

Report

In this paper, the authors study gauging of the fermion parity symmetry in the Kitaev chain. The paper presents a detailed study of the gauged fermion parity, and arrives at new results concerning the phases, vorticity and boundary effects. Before recommending the paper for publication, I suggest the authors to discuss the points raised below.

Requested changes

1) The authors have referred to gauge theory works in detail. But, still, it would be complementary to include a brief discussion of

Krauss LM, Wilczek F. Discrete gauge symmetry in continuum theories. Phys Rev Lett. 1989 Mar 13;62(11):1221-1223. doi: 10.1103/PhysRevLett.62.1221

This paper may help developing a more detailed (or slightly alternative) view of the transition from bulk to the boundary. (In gauge theories, boundary seldom appears, and this is one of the factors that make the present paper interesting.)

2) In Eq. (14), the coupling constant h needs be explained in terms of its origin. That “h” term appears as a gauge-fixing term. Is h a gauge-fixing parameter? What is then its physical relevance (other than causing vortices in Sec. 3.3 and Sec. 3.7)? Unless the emergent parameter h is clarified part of the analysis appears to result from parameter choices. For instance, authors take h=0 in Sec. 3.5 and Sec. 3.6 where the Higgs, deconfined and SPT phases are discussed. It would make the paper comprehensible and complete to include a discussion of h.

3) At the beginning of Sec. 3.4 the bosonic nature of the bulk needs be expanded somewhat. For example, what is the dynamical use/role of that “emergent fermion”?

4) In Fig. 3, the gauged (at the left?) and non-gauged (at the right?) Kitaev chains can be made more tractable.

5) In Sec. 4.1, Eq. (35), the coupling K multiplies the Gauss’ law. This means that Gauss’ law is imposed at high K, where at low K conservation laws are not guaranteed. Also role of K in the sense of effective field theory is not clear. It looks like small K means loss of Gauss’ law more than non-gauging (the authors propose). This point needs be clarified. (Is the anomaly in Sec. 5.2 connected to these features?) Speaking in general, the gentle gauging seems to involve somewhat ad hoc Hamiltonian in Eq.(35). The authors might want to reconsider this part in terms of discussions and analyses.

6) Fig. 5 can be extended from h=0 to larger h values to reveal the dependence of entropy on this gauge parameter.

7) I suggest the authors to have a full check of their equations and arguments. Things may be as simple as not defining hermiticity of γ ̃ in Eq. (2).

  • validity: high
  • significance: high
  • originality: high
  • clarity: good
  • formatting: good
  • grammar: good

Author:  Umberto Borla  on 2021-05-21  [id 1455]

(in reply to Report 2 on 2021-04-19)

We address below the comments of the Referee:

1) The authors have referred to gauge theory works in detail. But, still, it would be complementary to include a brief discussion of Krauss LM, Wilczek F. Discrete gauge symmetry in continuum theories. Phys Rev Lett. 1989 Mar 13;62(11):1221-1223. doi: 10.1103/PhysRevLett.62.1221 This paper may help developing a more detailed (or slightly alternative) view of the transition from bulk to the boundary. (In gauge theories, boundary seldom appears, and this is one of the factors that make the present paper interesting.)

We have added a few references when we first mention discrete gauge theory in the introduction, including the paper proposed by the referee. However, while the suggested paper points out some interesting general features of gauge theories, we do not see how the paper can "help developing a more detailed view of the transition from bulk to the boundary". To our understanding, boundary physics is never mentioned in the suggested paper.

Change in manuscript: added references 34-37.

2) In Eq. (14), the coupling constant h needs be explained in terms of its origin. That “h” term appears as a gauge-fixing term. Is h a gauge-fixing parameter? What is then its physical relevance (other than causing vortices in Sec. 3.3 and Sec. 3.7)? Unless the emergent parameter h is clarified part of the analysis appears to result from parameter choices. For instance, authors take h=0 in Sec. 3.5 and Sec. 3.6 where the Higgs, deconfined and SPT phases are discussed. It would make the paper comprehensible and complete to include a discussion of h.

The coupling constant $h$ is by no means a "gauge fixing" parameter. As explained in detail in sections 3.1 and 3.2, in the discrete $\mathbb{Z}_2$ gauge theory the $\sigma_x$ term plays a role analogous to the (exponential of) the electric field in the $U(1)$ electrodynamics. It is the simplest and most natural gauge invariant term that can be added to the Hamiltonian to include dynamics of the gauge fields (note that the $h$-term commutes with the Gauss operator in Eq.(10), making it gauge-invariant).

Perhaps the referee is referring to our conceptual motivation of this term: in the Lagrangian/spacetime formulation this term appears as an elementary Wilson loop (plaquette) with time-like links. In order to see that this corresponds to a $\sigma_x$ term in the Hamiltonian formalism, it is necessary to choose a gauge that fixes the link variables in the temporal direction before applying the usual classical-quantum correspondence. This is explained in the paragraph above Eq. (14), which aims to give a pedagogical explanation of our Hamiltonian to readers that are more used to the Lagrangian formalism.

Change in manuscript: Added a sentence to the manuscript (below Eq.(15)) to clarify this confusion: “We stress that although the $h$-term in Eq.(14) was motivated by fixing a gauge on temporal links, the term itself is manifestly gauge-invariant in this quantum lattice gauge theory---indeed, it commutes with the gauge constraint in Eq.(15). ”

3) At the beginning of Sec. 3.4 the bosonic nature of the bulk needs be expanded somewhat. For example, what is the dynamical use/role of that “emergent fermion”?

The concept of an emergent fermion is a standard occurrence in gauge theories: even though the (bulk) Hilbert space is manifestly bosonic, the low-energy quasiparticle excitations can be fermionic in nature. This is only possible for fractionalized particles, such that any local operator needs to create a pair -consistent with local operators being bosonic. The word ‘emergent’ refers to the fact that there is no local fermionic operator in this theory; this nomenclature is perhaps more common in the condensed matter literature, which is indeed the angle from which the present manuscript is written. We have now slightly expanded on this sentence and added references to previous works showing that this is a standard nomenclature.

Change in manuscript: we added a sentence “Similar to those encountered in higher-dimensional gauge theories, such emergent fermions can only be created in pairs, consistent with the fact that any local operator in the bulk of this theory is bosonic.”

4) In Fig. 3, the gauged (at the left?) and non-gauged (at the right?) Kitaev chains can be made more tractable.

We have clarified these issues by expanding the caption to Fig. 3.

Change in manuscript: The new caption reads: “The Kitaev chain (left) can be gauged by following the procedure outlined in section 3.2. The resulting Hamiltonian is identified with a TFIM (center) through a local transformation. Yet a different TFIM (right) can be obtained either as the Jordan-Wigner dual of the original Kitaev chain, or as the Kramers-Wannier dual of the aforementioned Ising model (i.e., of the gauged Kitaev chain).”

5) In Sec. 4.1, Eq. (35), the coupling K multiplies the Gauss’ law. This means that Gauss’ law is imposed at high K, where at low K conservation laws are not guaranteed. Also role of K in the sense of effective field theory is not clear. It looks like small K means loss of Gauss’ law more than non-gauging (the authors propose). This point needs be clarified. (Is the anomaly in Sec. 5.2 connected to these features?) Speaking in general, the gentle gauging seems to involve somewhat ad hoc Hamiltonian in Eq.(35). The authors might want to reconsider this part in terms of discussions and analyses.

Our intention in Sec. 4 is to to find a Hamiltonian that interpolates between the ordinary (ungauged) Kitaev model and the gauged Kitaev chain, and the model (35) does exactly that in a natural way. In the limit $K\rightarrow\infty$, we recover the gauged Kitaev model (with a hard Gauss law) investigated in Sec. 3. On the other hand, the $K \rightarrow 0$ limit corresponds precisely to the ordinary (ungauged) Kitaev chain. Indeed, the referee is correct in observing that in this limit, the Gauss law term disappears. However, at the same time, there is a term proportional to $1/K$ ensuring that at small $K$ the $\mathbb{Z}_2$ gauge degrees of freedom are frozen, i.e. $\sigma^z$ evaluates to $+1$ on all links. As a result, the fermionic part of the Hamiltonian reduces to the ordinary Kitaev chain. We already discuss this in detail in the manuscript below Eq.(35), and for this reason we have not made any changes. If the referee feels that any particular aspect is still unclear, we will be happy to consider any suggestions for improvement.

6) Fig. 5 can be extended from h=0 to larger h values to reveal the dependence of entropy on this gauge parameter.

Fig. 5 already shows the phase diagram for two different values of $h$ ($h=1$ and $h=2$). One further value ($h=5$) is shown in Fig. 6. For $h=0$ on the other hand the phase diagram is obtained analytically and displayed in Fig. 4. We feel that these four values give a sufficiently clear qualitative picture of how the phase boundaries depend on the electric coupling $h$. (Let us also emphasize again that $h$ is not a gauge parameter.)

Change in manuscript: in the caption Fig.5, we now explicitly refer to Fig.4 and Fig.6 for smaller/larger values of $h$, respectively.

7) I suggest the authors to have a full check of their equations and arguments. Things may be as simple as not defining hermiticity of $\tilde \gamma$ in Eq. (2).

We had left the hermiticity of $\tilde \gamma$ implicit since it is analogous to that of $\gamma$, but we are happy to state this more explicitly. We have further performed a full check of all our equations and arguments.

Change in manuscript: we added the hermiticity condition of $\tilde \gamma$.

---

## Round 3 · Referee Report · Anonymous (Referee 2) · 2021-5-24

Report

The authors have taken into account all the comments and suggestions in the referee report, and clarified their work accordingly. I recommend publication of this revised version in SciPost Physics.

---

## Round 3 · Referee Report · Paul Fendley (Referee 1) · 2021-5-25

Report

The changes are fine, and I recommend publication in SciPost.

---

## Round 3 · Author Response

We thank the editor for considering our paper for publication in SciPost.

We are grateful to both Referees for reading our paper and providing valuable feedback. In two replies we provide point-by-point answers to the comments of the Referees and the list of changes implemented in the present resubmission.

Best regards,
Umberto Borla, Ruben Verresen, Jeet Shah, Sergej Moroz

---

## Round 3 · List of Changes

• References: added references 34-37.

  • Figures: expanded the caption to Fig. 3

-Figures: in the caption of Fig.5, we now explicitly refer to Fig.4 and Fig.6 for smaller/larger values of h, respectively.

  • Section 1: Modified a sentence to “In particular, after gauging, the trivial phase of the Kitaev chain becomes a deconfined phase which for a positive chemical potential is stable to vortices, despite matter being massive, due to spontaneous translation symmetry-breaking giving rise to deconfined domain walls.”

  • Section 2: we added the hermiticity condition of $\tilde \gamma$.

-Section 3.2: Added the sentence “Moreover, let us mention that although the case h = 0 can mapped to a free-fermion problem, this requires a non-local mapping which can obscure some of the relevant physics, so we do not follow this route.”

-Section 3.2: Added the sentence: “We stress that although the $h$-term in Eq.(14) was motivated by fixing a gauge on temporal links, the term itself is manifestly gauge-invariant in this quantum lattice gauge theory - indeed, it commutes with the gauge constraint in Eq.(15). ”

  • Section 3.4: Added the sentence "Similar to those encountered in higher-dimensional gauge theories, such emergent fermions can only be created in pairs, consistent with the fact that any local operator in the bulk of this theory is bosonic."

---

## Editorial Decision

published